TRANSPARENT OPEN PROCESS ACCESS

# Endoplasmic reticulum stress actively suppresses hepatic molecular identity in damaged liver

Vanessa Dubois[1,†] , Céline Gheeraert[1,‡], Wouter Vankrunkelsven[2,‡] , Julie Dubois-Chevalier[1,§], Hélène Dehondt[1,§], Marie Bobowski-Gerard[1] , Manjula Vinod[1] , Francesco Paolo Zummo[1], Fabian Güiza[2] , Maheul Ploton[1], Emilie Dorchies[1], Laurent Pineau[1], Alexis Boulinguiez[1], Emmanuelle Vallez[1], Eloise Woitrain[1], Eric Baugé[1], Fanny Lalloyer[1], Christian Duhem[1], Nabil Rabhi[3], Ronald E van Kesteren[4] , Cheng-Ming Chiang[5], Steve Lancel[1], Hélène Duez[1], Jean-Sébastien Annicotte[3], Réjane Paumelle[1], Ilse Vanhorebeek[2], Greet Van den Berghe[2] , Bart Staels[1] , Philippe Lefebvre[1,¶] & Jérôme Eeckhoute[1,*,¶] 

## Abstract

Liver injury triggers adaptive remodeling of the hepatic transcriptome for repair/regeneration. We demonstrate that this involves particularly profound transcriptomic alterations where acute induction of genes involved in handling of endoplasmic reticulum stress (ERS) is accompanied by partial hepatic dedifferentiation. Importantly, widespread hepatic gene downregulation could not simply be ascribed to cofactor squelching secondary to ERS gene induction, but rather involves a combination of active repressive mechanisms. ERS acts through inhibition of the liver-identity (LIVER-ID) transcription factor (TF) network, initiated by rapid LIVER-ID TF protein loss. In addition, induction of the transcriptional repressor NFIL3 further contributes to LIVER-ID gene repression. Alteration to the liver TF repertoire translates into compromised activity of regulatory regions characterized by the densest co-recruitment of LIVER-ID TFs and decommissioning of BRD4 super-enhancers driving hepatic identity. While transient repression of the hepatic molecular identity is an intrinsic part of liver repair, sustained disequilibrium between the ERS and LIVER-ID transcriptional programs is linked to liver dysfunction as shown using mouse models of acute liver injury and livers from deceased human septic patients.

**Keywords** liver injury; NFIL3; PAR-bZIP; sepsis; super-enhancer
**Subject Categories** Chromatin, Transcription & Genomics

**Mol Syst Biol. (2020) 16: e9156**

## Introduction

The liver exerts instrumental homeostatic and detoxifying functions. This organ is also characterized by a unique capacity to regenerate (Abu Rmilah *et al*, 2019). Studies in mice subjected to liver regeneration subsequent to partial hepatectomy (PHx), a model of liver resection which is a frequent clinical practice to remove liver tumors (Liu *et al*, 2015a), have identified a role for endoplasmic reticulum stress (ERS) in this process (Liu *et al*, 2015b; Argemi *et al*, 2017). ERS, which results from the accumulation of unfolded/misfolded proteins in the ER lumen, triggers the unfolded protein response (UPR), aimed at restoring ER homeostasis. The UPR is controlled by three major ERS sensors, namely endoplasmic reticulum to nucleus signaling 1 (ERN1/IRE1), eukaryotic translation initiation factor 2 alpha kinase 3 (EIF2AK3/PERK), and activating transcription factor 6 (ATF6; Almanza *et al*, 2019). Signaling triggered by these sensors leads to activation of the Xbox-binding protein 1 (XBP1S), ATF4 and ATF6 transcription factors (TFs), and subsequent collaborative induction of ERS handling genes such as ER chaperones (Vihervaara *et al*, 2017; Almanza *et al*, 2019). Additional, non-transcriptional effects of the UPR involved in alleviating ERS also comprise the regulation of protein synthesis (mRNA translation) and degradation

1   Inserm, CHU Lille, Institut Pasteur de Lille, U1011-EGID, University of Lille, Lille, France
2   Clinical Division and Laboratory of Intensive Care Medicine, Department of Cellular and Molecular Medicine, KU Leuven, Leuven, Belgium
3   UMR 8199 - EGID, CNRS, Institut Pasteur de Lille, University of Lille, Lille, France
4   Center for Neurogenomics and Cognitive Research, Neuroscience Campus Amsterdam, VU University, Amsterdam, The Netherlands
5   Simmons Comprehensive Cancer Center, Departments of Biochemistry and Pharmacology, University of Texas Southwestern Medical Center, Dallas, TX, USA
*Corresponding author. Tel: +33 3 20 97 42 20; Fax: +33 3 20 97 42 19; E-mail: jerome.eeckhoute@inserm.fr
‡These authors contributed equally to this work as second authors
§These authors contributed equally to this work as third authors
¶These authors contributed equally to this work as senior authors
†Present address: Clinical and Experimental Endocrinology, Department of Chronic Diseases, Metabolism and Ageing (CHROMETA), KU Leuven, Leuven, Belgium

(Almanza *et al*, 2019). However, it has become clear that ERS bears functions beyond proteostasis *per se* (Hetz, 2012). For instance, liver regeneration upon PHx requires transient ERS to induce genes involved not only in proteostasis but also in acute-phase and DNA damage responses (Liu *et al*, 2015b; Argemi *et al*, 2017). Moreover, ERS has been linked to the (patho)physiological control of lipid and glucose metabolism in the liver (Rutkowski, 2019). The molecular mechanisms involved in ERS-mediated control of liver metabolic functions are still poorly defined. TFs activated by the UPR (including XBP1S, ATF4, DDIT3/CHOP, and ATF6) can directly bind to and modulate expression of specific metabolic genes. In addition, a handful of liver-enriched TFs display reduced expression or activity upon ERS through ill-defined mechanisms (Rutkowski, 2019). In general, while gene silencing substantially contributes to ERS-induced transcriptional regulation, the mechanisms accounting for these downregulations are seldom defined (Vihervaara *et al*, 2018; Almanza *et al*, 2019). Hence, ERS-induced transcriptional remodeling, especially gene downregulation, remains to be fully understood and its relevance toward liver pathophysiology to be better defined.

We and others have reported that hepatic gene transcription relies on networks of highly interconnected TFs (Kyrmizi *et al*, 2006), which are co-recruited to *cis*-regulatory modules (CRMs; Dubois-Chevalier *et al*, 2017), a conclusion corroborated by several studies in other systems reporting that extensive collaboration of TFs at CRMs is essential for their activities (e.g., Levo *et al*, 2017). These findings point to a requirement for a comprehensive assessment of changes in global TF expression/activity induced by (patho)physiological signals when aiming to define how transcriptional outputs are controlled. Here, we have used a functional genomics approach to characterize the molecular mechanisms responsible for hepatic gene transcriptional alterations triggered by ERS and to define how this relates to liver damage in mouse models of acute liver injury and livers from deceased human septic patients.

# Results

## Acute ERS recapitulates the loss of hepatic molecular identity observed following liver PHx through extensive and preferential repression of liver-identity genes

The hepatic response to ERS was defined using transcriptomic analysis of mouse primary hepatocytes (MPH) treated with thapsigargin for 4 h (Appendix Fig S1A). In addition to induction of the UPR (hereafter referred to as the ERS UP genes), we found a substantial fraction of regulated genes (~45%) downregulated upon ERS in MPH (ERS DOWN genes; Appendix Fig S1A). This regulatory pattern was conserved when analyzing the mouse liver transcriptome 8 h after a single intraperitoneal injection of tunicamycin, another ERS-inducing drug (Appendix Fig S1B and C; Arensdorf *et al*, 2013). Moreover, transcriptomic data mining using Short Time-series Expression Miner (STEM; Ernst *et al*, 2005; Rib *et al*, 2018), a tool defining the preferential dynamic patterns of gene expression, confirmed that transient ERS occurs upon PHx (Fig 1A and Appendix Fig S2; Reimold *et al*, 2000; Liu *et al*, 2015b). Strikingly, unlike ERS UP genes, which are mostly involved in housekeeping functions, ERS DOWN genes are linked to liver-specific

**Figure 1.  Acute ERS triggers massive transcriptomic alterations characterized by repression of LIVER-ID genes and loss of hepatic molecular identity.**

A   Top 2 significantly overrepresented expression patterns for ERS UP and ERS DOWN genes following PHx. Data show changes in the expression at 4, 10, 48 h, and 1 week after PHx (0 h) for genes comprised within each model profile of dynamic expression identified by STEM. The complete set of identified model profiles is provided in Appendix Fig S2.

B   Functional enrichment analyses were performed using ERS UP (*upper panels*) or ERS DOWN (*lower panels*) genes and the ToppGene Suite. KEGG Pathways with Bonferroni-corrected $P < 10^{-3}$ were considered, and similar terms were merged.

C   Bagplots showing the breadth of transcriptomic changes for the indicated datasets. Genes were positioned based on their basal expression levels in the control conditions and their FC ($Log_2$) in the indicated (patho)physiological context. The dark blue area is the "bag" (50% of the data points around the median, which is indicated by a red cross), while the light blue area delimits the "loop" (see Materials and Methods for details). Red dots are outliers.

D, E   Box plots showing normalized expression in liver (D) and liver-specificity index (E) of LIVER-ID genes, UBQ genes, and other genes. Liver-specificity index was calculated as the difference in normalized expression in liver (2 replicates) and mean of normalized expression in control tissues (2 replicates per tissue) using data from BioGPS (Table EV6) and is reported as $Log_2$. Box plots are composed of a box from the 25th to the 75th percentile with the median as a line and min to max as whiskers. One-way ANOVA with Welch's correction and Dunnett's modified Tukey–Kramer pairwise multiple comparison test was used to assess statistical significance, $*P < 0.05$.

F   Similar analyses as in (A) using LIVER-ID genes.

G, H   Box plots showing $Log_2$ FC ERS/control in MPH (3 independent experiments) (G) or mouse liver (3 mice per group) (H) for LIVER-ID genes, UBQ genes, and other genes. Box plots are composed of a box from the 25th to the 75th percentile with the median as a line and min to max as whiskers. One-way ANOVA with Welch's correction and Dunnett's modified Tukey–Kramer pairwise multiple comparison test was used to assess statistical significance, $*P < 0.05$.

I   Enrichment plots from gene set enrichment analyses (GSEA) performed using LIVER-ID genes as the gene set and transcriptomic changes induced by acute ERS in MPH (*upper panel*) or mouse liver (*lower panel*) as the ranked gene list. NES and FDR (as in all subsequent GSEA panels) are the normalized enrichment score and the false discovery rate provided by the GSEA software, respectively.

J, K   Genes repressed by ERS in MPH (3 independent experiments) (J) or mouse liver (3 mice per group) (K) were ranked based on their $Log_2$ FC ERS/control and divided into quartiles (increased repression from Q1 to Q4). The fraction of LIVER-ID genes in the 4 quartiles was defined and is displayed relative to that obtained for Q1 arbitrarily set to 1. Chi-square test with BH correction for multiple testing was used to assess statistical significance, $*P < 0.05$.

L, M   RT–qPCR analyses of selected ERS UP and LIVER-ID genes monitoring expression changes induced by acute ERS in MPH (3–9 independent experiments) (L) or mouse liver (5–7 mice per group) (M). The bar graphs show means ± SD (standard deviations). One-sample *t*-test with BH correction for multiple testing was used to determine whether the mean $Log_2$ FC ERS/control is statistically different from 0, $*P < 0.05$. Panel (L) is also displayed in Appendix Fig S3G.

N   Heatmaps showing normalized expression of ERS UP and ERS DOWN genes from MPH (*upper panel*) or mouse liver (*lower panel*) in mouse liver at the indicated stages of development.

O   Average expression of ERS DOWN genes from MPH in single cells from the hepatobiliary lineage. See Materials and Methods together with Appendix Fig S3I for details regarding data processing. The hepatoblast-to-hepatocyte and hepatoblast-to-cholangiocyte differentiation paths are indicated with arrows.

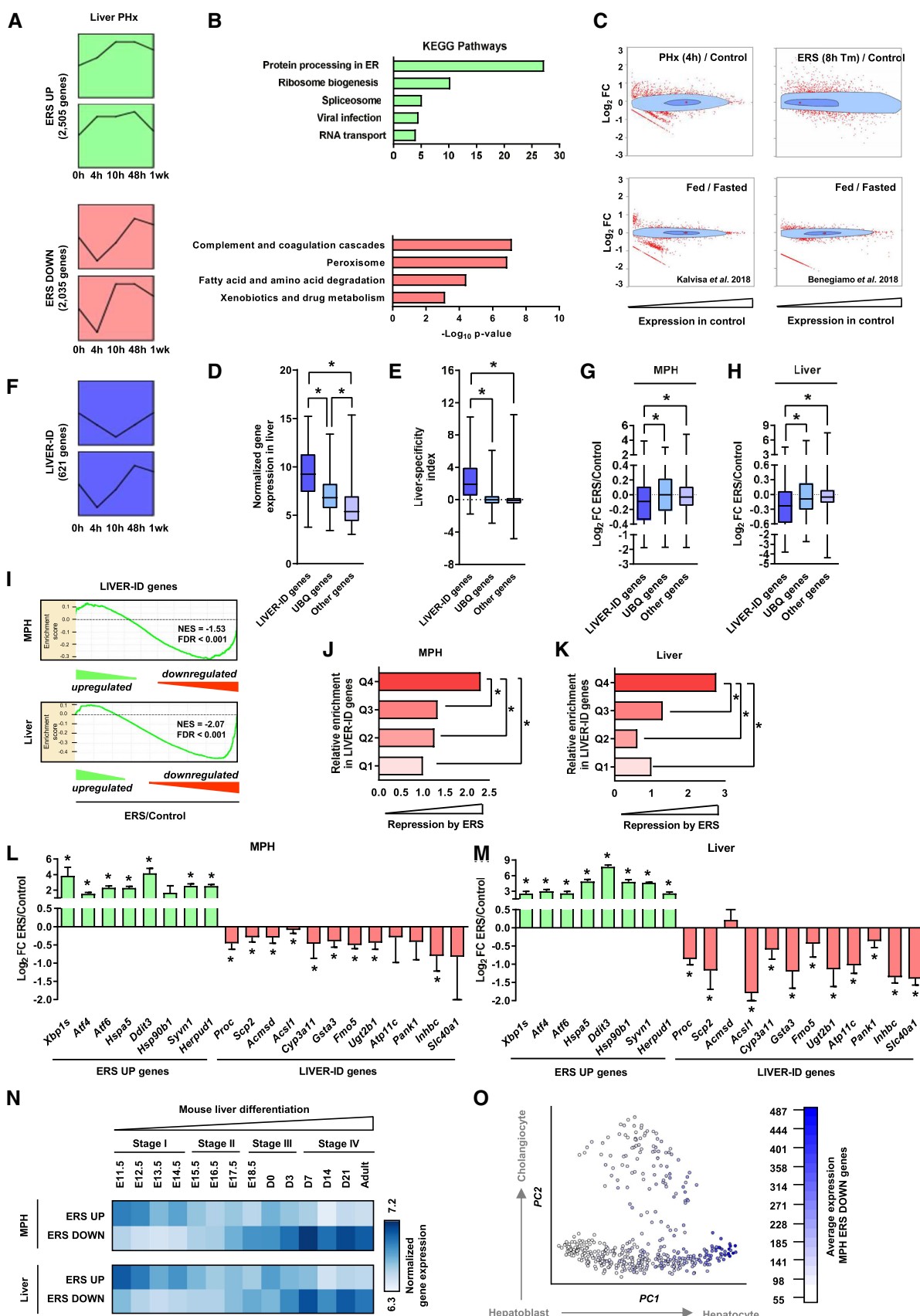

**Figure 1.**

functions (e.g., coagulation, xenobiotic drug metabolism; Fig 1B). Concomitant to transient ERS, mouse liver regeneration following PHx has been linked to transient decrease in metabolic gene expression (White et al, 2005; Argemi et al, 2017). We found this was related to ERS DOWN genes being transiently downregulated upon liver PHx (Fig 1A). As the transcriptome of cells has been proposed to be ruled by ecosystem-like equilibriums where resources required to induce novel programs are used at the expense of established ones (Silveira & Bilodeau, 2018), we monitored the extent of transcriptomic alterations triggered by PHx or chemically induced ERS compared with physiological transcriptomic changes unrelated to liver injury, i.e., triggered by fasting to feeding transition in mice (Benegiamo et al, 2018; Kalvisa et al, 2018). Bagplots (bivariate boxplots showing fold changes in expression relative to baseline mouse liver gene expression levels) revealed that downregulation of the hepatic program upon PHx and ERS was associated with more complex and widespread transcriptomic alterations including a greater induction of genes expressed at low/moderate levels in the healthy mouse liver (Fig 1C).

To further characterize the impact of PHx and ERS on the hepatic transcriptional program, we defined cell identity genes, i.e., master liver transcriptional regulators and effector genes. Identity genes establish/maintain tissue-specific functions and distinguish themselves by broad H3K4me3 domains encompassing the transcription start site, a feature functionally related to high transcriptional expression and consistency (Benayoun et al, 2014). We defined liver-identity (LIVER-ID) genes as those displaying this epigenetic feature preferentially in liver (Appendix Fig S3A and Table EV1). We verified that LIVER-ID genes displayed expression levels which are higher (Fig 1D), liver-specific (Fig 1E), and linked to hepatic functions when compared to non-LIVER-ID genes, i.e., ubiquitously labeled with broad H3K4me3 (UBQ genes) or lacking liver broad H3K4me3 (Other genes; Appendix Fig S3B). LIVER-ID genes were transiently downregulated upon liver PHx (Fig 1F) as well as upon ERS both in vitro in MPH and in vivo in mouse liver

(Fig 1G and H, Appendix Fig S3C and D). LIVER-ID gene repression was specific and not linked to their high expression levels which could make them more prone to repression, since ERS-mediated repression did not correlate with basal hepatic gene expression (Appendix Fig S3E). Note that while microarray-based transcriptomic analyses reliably define fold changes, this technology under-estimates their magnitude (Dallas et al, 2005). This notion should be taken into account when interpreting microarray-based results throughout the study. Remarkably, LIVER-ID genes were enriched among genes which were most strongly repressed by ERS (Fig 1I–K). Reciprocal induction of ERS UP genes and downregulation of LIVER-ID genes was validated using RT–qPCR in MPH (Fig 1L, Appendix Fig S3F and G) and mouse liver (Fig 1M and Appendix Fig S3F). Interestingly, ERS DOWN genes correspond to genes induced during hepatic differentiation, based on whole liver (Fig 1N and Appendix Fig S3H; Li et al, 2009) or hepatobiliary single-cell (Fig 1O and Appendix Fig S3I and J; Yang et al, 2017) transcriptomic data obtained at different developmental stages.

Altogether, these data point to loss of hepatic molecular identity upon liver PHx, which can be recapitulated by acute ERS acting as a widespread repressor of the liver transcriptional program.

## Acute ERS triggers a global loss of activity of the LIVER-ID TF network and its densely co-bound CRMs

To define how liver molecular identity loss is induced by ERS at the transcriptional regulatory level, we monitored changes in CRM activities in MPH using alterations to H3K27 acetylation (H3K27ac) levels as a surrogate. H3K27ac ChIP-seq assays identified regions with increased (62%) or decreased (38%) H3K27ac signal intensities (denoted H3K27ac UP or H3K27ac DOWN), respectively (Appendix Fig S4A and Dataset EV1). Genes linked to H3K27ac UP regions were significantly enriched in ERS UP genes, while ERS

**Figure 2. Acute ERS compromises LIVER-TF expression and activities of their densely co-bound hepatic CRMs.**

A   Comparison of transcriptomics (data from three independent experiments) and H3K27ac ChIP-seq (data from three independent experiments) from MPH. Genes were assigned to H3K27ac regions as described in Materials and Methods. The number of ERS DOWN genes is indicated relative to the number of ERS UP genes for the three categories of H3K27ac regions. Fisher's exact test with BH correction for multiple testing was used to assess statistical significance, *P < 0.05, #P < 0.05.

B   Similar analyses to (A). The number of LIVER-ID and UBQ genes is indicated relative to the number of other genes for the three categories of H3K27ac regions. The H3K27ac ChIP-seq data were obtained from three independent MPH experiments. Chi-square test with BH correction for multiple testing was used to assess statistical significance, #P < 0.05.

C   Multidimensional scaling (MDS) was performed as described in Materials and Methods, and transcriptional regulator co-recruitment was depicted using density plots for regions with increased (a), decreased (b) or unchanged (c) H3K27ac levels in MPH upon acute ERS. The boxed area (d) represents a subset of transcriptional regulators (TR) with a high degree of co-binding in H3K27ac DOWN regions. LIVER-ID TFs are depicted in red.

D   Heatmaps showing the percentage of enhancers overlapping H3K27ac down regions. Enhancers were first split into those bound or not by a given LIVER-ID TF (right) and then based on co-binding of additional LIVER-ID TFs (defining three subgroups with 0–2, 3–5, or 6–8 co-bound additional LIVER-ID TFs).

E   Enrichment plots from GSEA performed using LIVER-ID TFs as the gene set and transcriptomic changes induced by acute ERS in MPH (upper panel) or mouse liver (lower panel) as the ranked gene list.

F   RT–qPCR analyses of selected LIVER-ID TFs monitoring expression changes induced by acute ERS in MPH (4 to 7 independent experiments). The bar graph shows means ± SD (standard deviations). One-sample t-test with BH correction for multiple testing was used to determine whether the mean Log2 FC ERS/control is statistically different from 0, *P < 0.05. This graph is also displayed in Appendix Fig S6E.

G   RT–qPCR analyses of selected ERS UP and LIVER-ID TF genes monitoring expression changes induced by 4 h ERS in MPH pre-treated or not for 30 min with 5 mM PBA (three independent experiments). Mean Log2 FC ERS/control is shown. The bar graph shows means ± SD (standard deviations). Two-way ANOVA with Bonferroni's post hoc test was used to assess statistical significance, *P < 0.05.

H, I   Nuclear extracts (H) and chromatin fractions (I) from MPH were subjected to Western blot with antibodies against HNF4A, NR1H4/FXR, FOXA2/HNF3B, or DDIT3/CHOP. LMNA or histone H3 was used as loading control. Results obtained from 3 independent biological replicates are shown. See Appendix Fig S6H and Ploton et al (2018) for antibody validation.

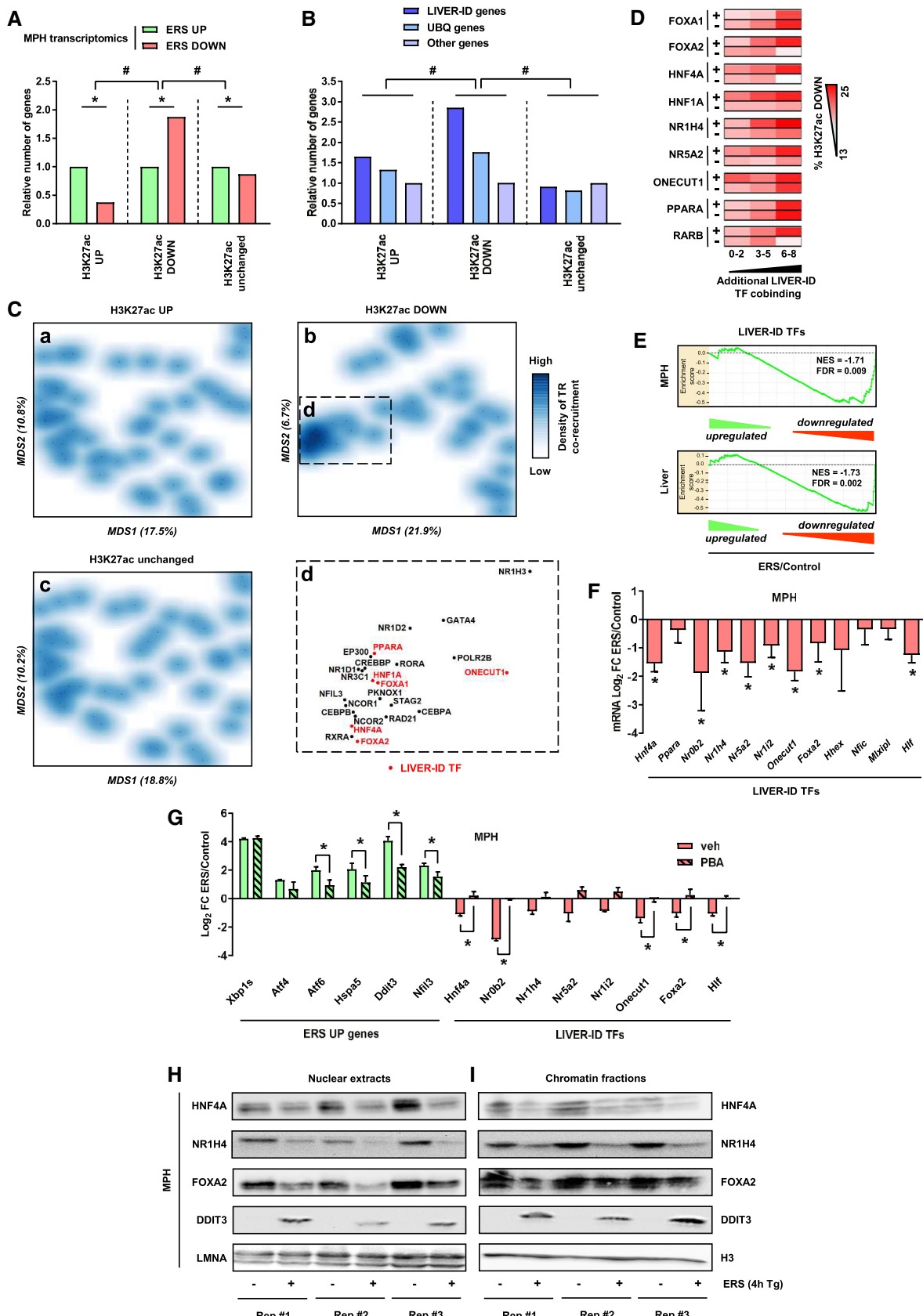

**Figure 2.**

DOWN genes were more strongly linked to H3K27ac DOWN regions (Fig 2A). In line, LIVER-ID genes were most significantly linked to H3K27ac DOWN regions (Fig 2B).

Locus overlap analysis (LOLA; Sheffield & Bock, 2016) was next used to compare genomic localization of H3K27ac regions with the chromatin-binding sites (cistromes) of mouse TFs (657 cistromes) from the Gene Transcription Regulation Database (GTRD; Yevshin *et al*, 2017). We found that 81% of the mouse hepatic TF cistromes comprised in this database (61 out of 75 cistromes) belonged to the cluster displaying the strongest overlap with H3K27ac DOWN regions (Appendix Fig S4B and Table EV2), suggesting these regions might be characterized by dense hepatic TF co-recruitment. In line, monitoring the co-recruitment patterns of 47 transcriptional regulators in mouse liver (Dubois-Chevalier *et al*, 2017) revealed that H3K27ac DOWN regions were uniquely characterized by a node of highly co-recruited transcriptional regulators, which was not found for H3K27ac UP or unchanged regions (Fig 2C). Moreover, several of these regulators were defined as LIVER-ID TFs (Fig 2C, panel D), i.e., TFs comprised within the previously described LIVER-ID gene list (Appendix Fig S5 and Table EV1; 43 TFs).

Additional analyses of hepatic enhancers indicated that the extent of LIVER-ID TF co-binding positively associated with their chance of being inactivated by ERS, as judged through their overlap with H3K27ac DOWN regions (Fig 2D). Importantly, this could not be attributed to any single LIVER-ID TF. Indeed, this pattern was observed for each individual LIVER-ID TF, when focusing on its specific set of bound CRMs. Additionally, lack of several individual LIVER-ID TFs was linked to reduced propensity for ERS-mediated repression. Thus, deficiency in activity of a single LIVER-ID TF could not explain ERS-induced loss of H3K27ac at mouse CRM but rather pointed to ERS-mediated inactivation being linked to a concomitant impaired activity of several LIVER-ID TFs (Fig 2D). This led us to investigate whether ERS is linked to a coordinated and global loss of master hepatic TF activities. In line with this hypothesis, we found that LIVER-ID TFs mostly belonged to ERS DOWN genes (Fig 2E and Appendix Fig S6A and B). Similar conclusions were reached when defining LIVER-ID TFs based on high and specific expression in the human liver (D'Alessio Ana *et al*, 2015) or on reconstructed gene regulatory networks (Zhou *et al*, 2017),

which largely overlap with our epigenetically defined LIVER-ID TF list (Appendix Fig S6C and D), hence supporting the robustness of our findings. RT–qPCR assays confirmed ERS-mediated downregulation of LIVER-ID TFs in mouse AML12 hepatocytes, MPH, and mouse liver (Fig 2F and Appendix Fig S6E–G). LIVER-ID TF gene downregulation in MPH was blunted by pre-treating the cells with the chemical chaperone 4-phenylbutyrate (PBA), which, as expected, alleviated ERS response as judged through lower induction of ERS UP genes (Fig 2G). The diminished LIVER-ID TF gene expression induced by ERS was accompanied by loss of LIVER-ID TF activity as evidenced by a drastic decrease in both nuclear and chromatin-bound levels of HNF4A, NR1H4/FXR, and FOXA2/HNF3B (Fig 2H and I), which have well-established hepatic functions and were used as examples in these analyses.

These data indicate that ERS triggers a global impairment of LIVER-ID TF expression/activities. Among LIVER-ID TF with decreased expression, HLF (Fig 2F and Appendix Fig S6E and F) belongs to the PAR-bZIP family together with TEF, DBP, and NFIL3. While HLF, TEF, and DBP behave as transcriptional activators, repressive functions have been ascribed to NFIL3, thereby establishing a balance in the transcriptional regulation of shared target genes (e.g., Mitsui *et al*, 2001). Recently, NFIL3 has been found to be induced upon ERS in mouse pancreatic islets (Ohta *et al*, 2017). Interestingly, acute hepatic ERS triggers a switch in the expression profile of the PAR-bZIP TF family members including strong induction of *Nfil3* levels (Fig 3A and B, Appendix Fig S7A–D) and chromatin binding (Fig 3C). This switch in the expression of the PAR-bZIP TF family members was also observed upon liver PHx (Appendix Fig S7E). Transcriptomic analyses of the liver of *Nfil3*$^{-/-}$ (NFIL3 KO) mice subjected to ERS (Appendix Fig S8) revealed that repression of LIVER-ID gene expression was blunted in NFIL3 KO mice, as illustrated by GSEA showing enrichment of LIVER-ID genes in NFIL3 KO compared with WT livers subjected to acute ERS (Fig 3D). Further investigation of the genes contributing the most to this enrichment, i.e., LIVER-ID genes less efficiently repressed by ERS in NFIL3 KO compared with WT mice (Table EV3), revealed "Drug metabolism and cytochrome P450" as the main pathway (Fig 3D). Several of these genes are involved in xenobiotic metabolism (*Gsta3, Adh1, Cyp3a11, Fmo5,* and *Ugt2b1*), a liver function

**Figure 3. Induction of NFIL3 by acute ERS contributes to repression of genes involved in xenobiotic metabolism.**

A  RT–qPCR analyses of *Hlf, Tef, Dbp*, and *Nfil3* expression monitoring changes induced by acute ERS in MPH (3–5 independent experiments) (*left panel*) or mouse liver (five mice per group) (*right panel*). The bar graphs show means ± SD (standard deviations). One-sample *t*-test with BH correction for multiple testing was used to determine whether the mean Log$_2$ FC ERS/control is statistically different from 0, *P < 0.05.

B  Total protein extracts from MPH (*left panel*) or mouse liver (*right panel*) were subjected to Western blot with an antibody against NFIL3. ACTB was used as loading control. See Appendix Fig S7B for antibody validation.

C  Chromatin fractions from MPH (*upper panel*) or mouse liver (*lower panel*) were subjected to Western blot with an antibody against NFIL3. Histone H3 was used as loading control.

D  Enrichment scores from GSEA performed using LIVER-ID genes repressed by acute ERS in MPH (MPH ERS DOWN) as the gene set and liver transcriptomic changes induced by acute ERS and/or deletion of *Nfil3* (NFIL3 KO) as the ranked gene lists were integrated and corrected for multiple testing using the BubbleGUM tool. For the NFIL3 KO ERS vs WT ERS comparison, the Core Enrichment genes (i.e., the subset of genes that contributes most to the enrichment result) were subjected to functional enrichment analyses using the ToppGene Suite. The top ranked KEGG Pathway with its Bonferroni-corrected *P*-value is shown.

E  Box plots showing mRNA expression for 5 genes from the Core Enrichment from D involved in xenobiotic metabolism issued from the transcriptomic analyses. Shown are Log$_2$ FC relative to the mean normalized expression in the WT control group (five mice per group). Box plots are composed of a box from the 25$^{th}$ to the 75$^{th}$ percentile with the median as a line and min to max as whiskers. Two-way ANOVA with Bonferroni's post hoc test was used to assess statistical significance, *P < 0.05.

F  The Integrated Genome Browser (IGB) was used to visualize ChIP-seq profiles for NFIL3 (green) and several LIVER-ID TFs (red) in the mouse liver at the *Gsta3* gene locus. Levels of H3K27ac in MPH and cells from the non-parenchymal fraction (NPC) are shown in blue. The grey bar indicates the position of a BRD4 SE.

previously defined as being critically regulated by the PAR-bZIP TF family (Gachon *et al*, 2006). Additional cytochrome P450-related liver functions are also less repressed by ERS in NFIL3 KO mice (Table EV3). Blunted ERS-mediated repression in NFIL3 KO mouse

livers of specific genes related to xenobiotic metabolism is shown in Fig 3E. In addition to the features defining the highest sensitivity to ERS-mediated repression (i.e., dense LIVER-ID TF co-recruitment and, as described hereafter, overlap with BRD4 SE), these genes are

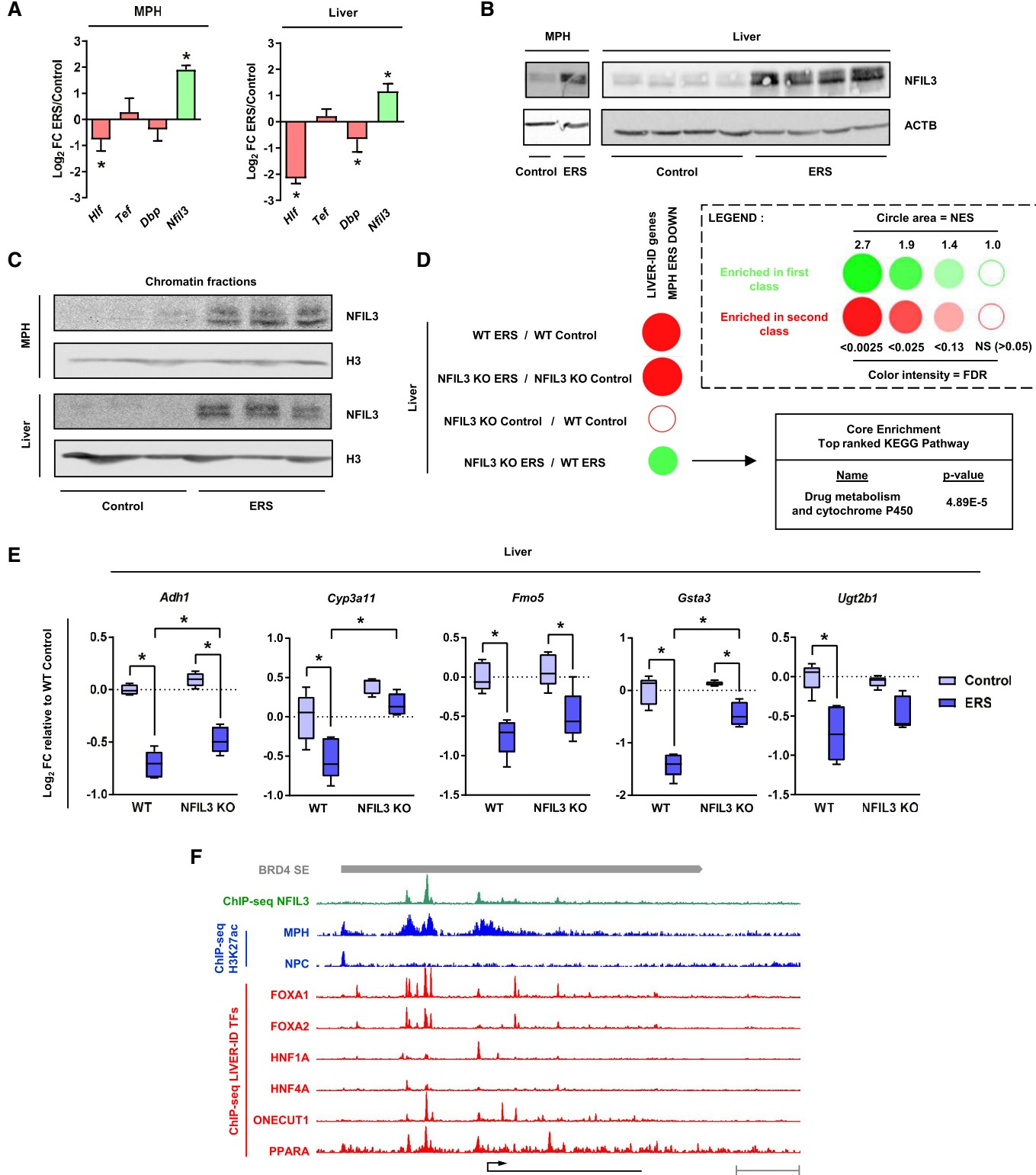

Figure 3.

bound by NFIL3 at CRMs which are specifically active in hepatocytes when compared to the non-parenchymal cells (NPC; Fig 3F and Appendix Fig S9).

Altogether, these data reveal that acute ERS profoundly remodels the liver TF repertoire where a global loss of LIVER-ID TF expression is reinforced, within the PAR-bZIP TF family, by induction of the transcriptional repressor NFIL3.

### Acute ERS triggers decommissioning of BRD4 at super-enhancers (SE) and preferentially represses SE-associated genes

We next investigated how compromised LIVER-ID TF expression/ activities translate into loss of the hepatic transcriptional program. TFs activate target gene expression through recruitment of transcriptional coactivators. Among those, BRD4 has been identified as crucial to establish and maintain transcriptomic cellular identity (Di Micco et al, 2014; Lee et al, 2017). BRD4 recruitment to CRMs requires EP300-mediated protein acetylation, including acetylation of H3 at K27 (Roe et al, 2015). Considering that LIVER-ID TFs target EP300 at liver CRMs to catalyze H3K27ac (Thakur et al, 2019), through their interaction with EP300 (Appendix Fig S10A; Eeck-houte et al, 2004; Kemper et al, 2009; von Meyenn et al, 2013), we hypothesized that the ERS-mediated decrease in LIVER-ID TFs and H3K27ac might culminate into compromised BRD4 recruitment at CRMs. Using ChIP-qPCR, we found that ERS decreased BRD4 binding at H3K27ac DOWN CRMs in MPH and mouse liver (Fig 4A and Appendix Fig S10B). This decrease could not be ascribed to reduced BRD4 expression levels upon ERS (Fig 4B and Appendix Fig S10C).

To verify the importance of BRD4 loss with regard to expression of LIVER-ID genes, MPH were treated with the BRD4 inhibitors JQ1, which impedes recognition of acetylated proteins by the BRD4 bromodomain, and MZ1, which targets BRD4 for degradation (Zengerle et al, 2015; Fig 4B). Both treatments severely compromised basal expression of LIVER-ID genes, which could moreover not be further repressed by ERS (Fig 4C and Appendix Fig S11). The decreased LIVER-ID gene expression observed upon BRD4 inhibition was not kinetically preceded by decreased protein expression of the LIVER-ID TFs HNF4A, NR1H4/FXR, and FOXA2/HNF3B (Appendix Fig S12), indicating it was most likely a direct

consequence of deficient BRD4-mediated gene transcription. Activation of canonical ERS UP genes was also blunted, although these genes appeared less sensitive to BRD4 inhibition (Fig 4C and Appendix Fig S11). MPH treatment with C646, an inhibitor of the EP300 histone acetyltransferase, also reduced basal expression of LIVER-ID genes and blocked further repression by ERS (Appendix Fig S13), while treatment with the histone deacetylase (HDAC) inhibitor trichostatin A impeded their repression by ERS (Appendix Fig S14).

BRD4 control of cell identity is linked to its recruitment at CRMs densely co-bound by master TFs, organized in clusters defining so called super-enhancers (SE) (Whyte et al, 2013). BRD4 SE [defined using mouse liver BRD4 ChIP-seq data (Kim et al, 2018)] largely overlapped LIVER-ID domains (i.e., liver preferential broad H3K4me3 regions as defined above; Appendix Fig S15A). Interestingly, LIVER-ID domains overlapping BRD4 SE showed stronger binding of EP300 and LIVER-ID TFs (Appendix Fig S15B) together with stronger H3K27ac basal levels in MPH, which displayed a more pronounced decrease upon acute ERS (Fig 4D). Accordingly, LIVER-ID domains overlapping BRD4 SE showed enrichment for H3K27ac DOWN regions (Appendix Fig S15C) and their associated genes (Appendix Fig S15D). In line, genes associated with both LIVER-ID domains and BRD4 SE showed the highest basal and most liver-specific expression (Appendix Fig S15E). Importantly, these genes were more strongly downregulated by ERS (Fig 4E).

Altogether, these data point to BRD4 SE decommissioning as central to LIVER-ID gene expression loss upon acute ERS in hepatocytes. In this context, BRD4 SE define a subset of LIVER-ID genes with greatest sensitivity to acute ERS-mediated repression.

### Loss of liver-identity is initiated by a rapid decrease in LIVER-ID TF protein levels upon acute ERS

Kinetic experiments in MPH indicated that the decrease in LIVER-ID TF expression triggered by ERS precedes that of non-TF LIVER-ID genes (Fig 5A), consistent with impairment of the hepatic TF network driving subsequent loss of liver CRM activities and target gene expression. LIVER-ID TFs are organized as an

---

**Figure 4. Acute ERS triggers hepatic SE decommissioning through impaired recruitment of the cell identity maintenance cofactor BRD4.**

A  BRD4 occupancy (*left panels*) and H3K27ac levels (*right panels*) at regulatory regions associated with ERS UP or LIVER-ID genes (8 regions at ERS gene loci and 10 regions at LIVER-ID gene loci, depicted in Appendix Fig S10B and listed in Table EV5) were assessed by ChIP-qPCR in MPH (10 independent experiments) or mouse liver (10 mice per group) to define changes induced by acute ERS. Box plots are composed of a box from the 25th to the 75th percentile with the median as a line and min to max as whiskers. One-sample *t*-test with BH correction for multiple testing was used to determine whether the mean Log$_2$ FC ERS/control is statistically different from 0, *$P < 0.05$.

B  *Left panel*, BRD4 mRNA (4 independent experiments) or protein expression levels (5 independent experiments; densitometric quantification of Fig 4B and Appendix Fig S10C) in MPH subjected to acute ERS. The bar graphs show means ± SD (standard deviations). Student's *t*-test was used to assess statistical significance. *Right panel*, Total protein extracts from MPH pre-treated for 3 h with 0.01 μM MZ1 followed by addition of 1 μM thapsigargin (ERS) for 4 h were subjected to Western blot with an antibody against the N-terminus of BRD4 (Wu et al, 2006). LMNA was used as loading control.

C  Heatmaps showing Log$_2$ FC (relative to the DMSO-control condition) for 5 ERS UP and LIVER-ID genes issued from RT–qPCR analyses (Appendix Fig S11) of MPH pre-treated with 500 nM JQ1 (*left*) or 0.01 μM MZ1 (*right*) followed by addition of 1 μM thapsigargin (ERS) for 4 h (3–6 independent experiments). Two-way ANOVA with Bonferroni's post hoc test was used to assess statistical significance.

D  Heatmaps showing average H3K27ac ChIP-seq signals in MPH at LIVER-ID domains overlapping (+) or not (-) with BRD4 SE. The arrow indicates the position of gene transcriptional start sites.

E  Box plots showing Log$_2$ FC ERS/control in MPH (three independent experiments) (*left panel*) or mouse liver (three mice per group) (*right panel*) for genes associated with LIVER-ID + BRD4 SE or LIVER-ID - BRD4 SE, which are listed in Table EV1. Box plots are composed of a box from the 25th to the 75th percentile with the median as a line and min to max as whiskers. Student's *t*-test was used to assess statistical significance, *$P < 0.05$.

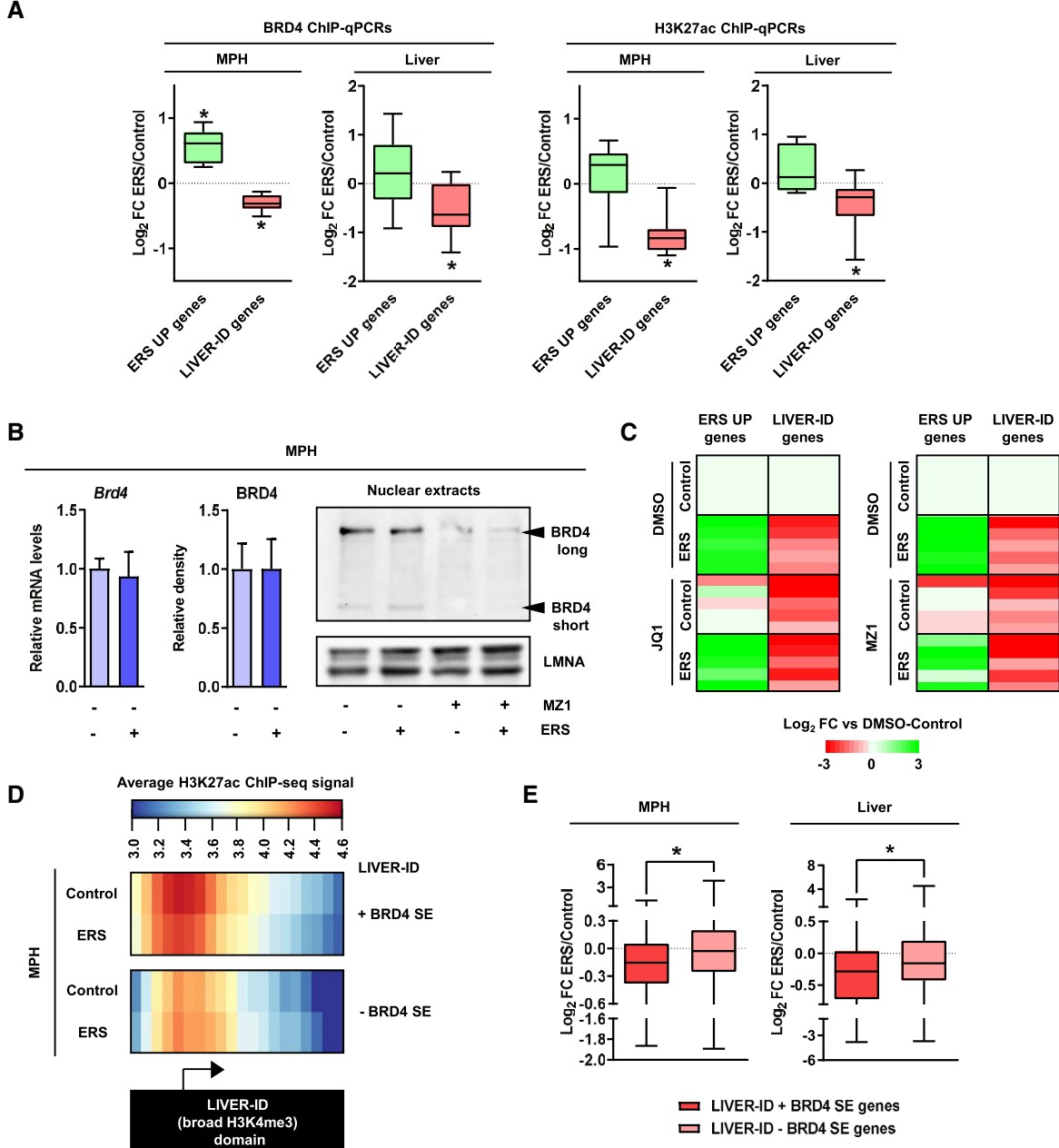

Figure 4.

interdependent transcriptional network involving auto- and cross-regulatory loops (Kyrmizi et al, 2006) at BRD4 SE (Appendix Fig S16). Therefore, we reasoned that any alteration to LIVER-ID TF protein levels could serve as an initial trigger which would secondarily be amplified by decreased expression of their encoding genes. Interestingly, further analyses of the kinetics of LIVER-ID TF loss of expression upon acute ERS revealed that the decrease in protein levels of HNF4A, NR1H4/FXR, and FOXA2/HNF3B was already effective 1 h after induction of ERS, at a time when mRNA levels are still unchanged (Fig 5B and C, Appendix Fig S17). While promoting translation of specific ERS-induced genes such as Atf4, ERS is also known to trigger global translation inhibition through the EIF2AK3/PERK pathway (Almanza et al, 2019).

We therefore hypothesized that ERS may modulate LIVER-ID TF activities by inhibiting their translation. In line with this hypothesis, treatment of MPH with the EIF2AK3/PERK signaling inhibitor ISRIB dampened loss of NR1H4 after 1 h of ERS (Fig 5D and E). Since translational inhibition alone could not account for acute ERS-induced LIVER-ID TF loss, and since ERS may modulate TF activities by inducing their degradation as shown for FOXO1 (Zhou et al, 2011) and CREB3L3 (Wei et al, 2018), we investigated a role for proteasomal degradation in LIVER-ID TF loss. We observed that treatment with the proteasome inhibitor MG132 blunted repression of LIVER-ID TFs, especially that of HNF4A, in MPH subjected to 1 h of ERS (Fig 5F and G, Appendix Fig S18A). Since proteasomal degradation of HNF4A can be induced by the

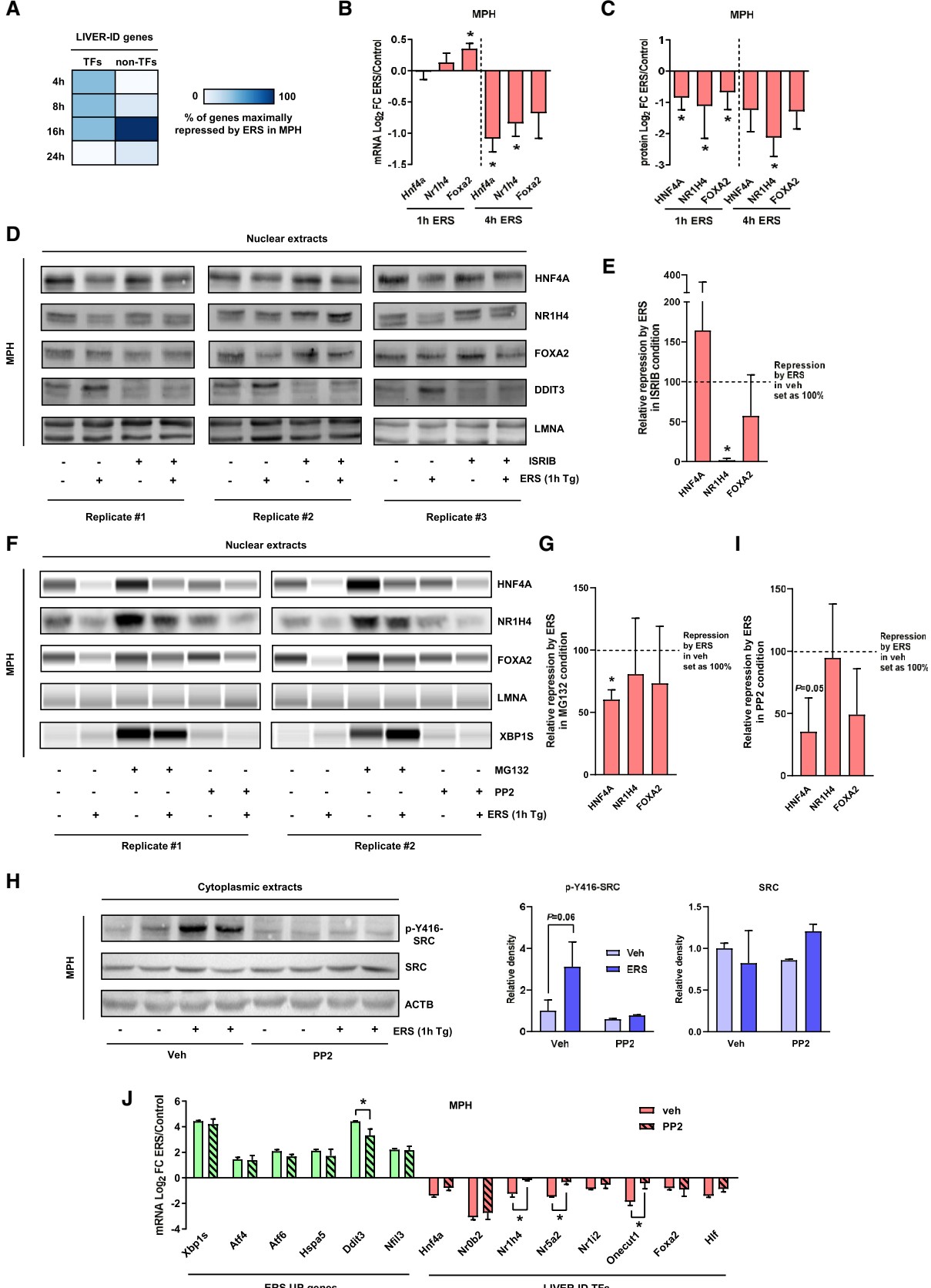

Figure 5.

**Figure 5.  Loss of LIVER-ID TF protein expression is a primary event upon acute ERS.**

A    RT–qPCR analyses of 21 LIVER-ID genes (12 TFs and 9 non-TFs, listed in Table EV5) in MPH which were treated with 1 μM thapsigargin (ERS) for 4, 8, 16, or 24 h (four independent experiments). For each gene, the timepoint showing maximal repression by ERS was recorded. The heatmap depicts the percentage of genes maximally repressed by ERS at each timepoint.

B    RT–qPCR analyses of *Hnf4a*, *Nr1h4*, and *Foxa2* expression in MPH treated with vehicle (control) or 1 μM thapsigargin (ERS) for 1 or 4 h (four independent experiments). The bar graph shows means $\pm$ SD (standard deviations). One-sample *t*-test with BH correction for multiple testing was used to determine whether the mean $\text{Log}_2$ FC ERS/control is statistically different from 0, *P < 0.05.

C    Densitometric quantification of the protein expression data shown in Figs 5D and 5F, Appendix Fig S18A and B for the 1-h timepoint (average of 9 biological replicates) and Fig 2H for the 4-h timepoint (average of 3 biological replicates). The bar graph shows means $\pm$ SD (standard deviations). One-sample *t*-test with BH correction for multiple testing was used to determine whether the mean Log2 FC ERS/control is statistically different from 0, *P < 0.05.

D    Nuclear extracts from MPH pre-treated for 30 min with 1 μM ISRIB followed by addition of 1 μM thapsigargin (ERS) for 1 h were subjected to Western blot with antibodies against HNF4A, NR1H4, FOXA2, or DDIT3. LMNA was used as loading control. Results obtained from three independent biological replicates are shown.

E    Densitometric quantification of the protein expression data shown in panel (D). Repression by ERS in the ISRIB condition (average of three biological replicates) is shown relative to repression by ERS in vehicle condition. The bar graph shows means $\pm$ SD (standard deviations). One-sample *t*-test with BH correction for multiple testing was used to determine whether the mean relative repression is statistically different from 100%, *P < 0.05.

F    Nuclear extracts from MPH pre-treated for 30 min with 10 μM MG132 or 10 μM PP2 followed by addition of 1 μM thapsigargin (ERS) for 1 h were subjected to Western blot or Simple Western immunoassay with antibodies against HNF4A, NR1H4, FOXA2, or XBP1S. LMNA was used as loading control. Results obtained from 2 independent biological replicates are shown. Additional replicates are shown in Appendix Fig S18A and B.

G    Densitometric quantification of the protein expression data shown in panel (F) and Appendix Fig S18A. Repression by ERS in MG132 condition (average of 4 biological replicates) is shown relative to repression by ERS in vehicle condition. The bar graph shows means $\pm$ SD (standard deviations). One-sample *t*-test with BH correction for multiple testing was used to determine whether the mean relative repression is statistically different from 100%, *P < 0.05.

H    (*Left*) Cytoplasmic extracts issued from the MPH used in panel (F) (2 biological replicates) were subjected to Western blot with antibodies against *P*-Y416-SRC or total SRC. ACTB was used as loading control. (*Right*) Densitometric quantification of the protein expression data. The bar graphs show means $\pm$ SD (standard deviations). Two-way ANOVA with Bonferroni's post-hoc test was used to assess statistical significance.

I    Densitometric quantification of the protein expression data shown in panel (F) and Appendix Fig S18B. Repression by ERS in the PP2 condition (average of four biological replicates) is shown relative to repression by ERS in vehicle condition. The bar graph shows means $\pm$ SD (standard deviations). One-sample *t*-test with BH correction for multiple testing was used to determine whether the mean relative repression is statistically different from 100%.

J    RT–qPCR analyses of selected ERS UP and LIVER-ID TF genes monitoring expression changes induced by 4-h ERS in MPH pre-treated or not for 30 min with 10 μM PP2 (3 independent experiments). Mean $\text{Log}_2$ FC ERS/control is shown. The bar graph shows means $\pm$ SD (standard deviations). Two-way ANOVA with Bonferroni's post hoc test was used to assess statistical significance, *P < 0.05.

SRC kinase (Chellappa *et al*, 2012) following liver PHx (Huck *et al*, 2019) and since ERS has been reported to activate SRC through its interaction with ERN1/IRE1a in HeLa cells (Tsai *et al*, 2018), we monitored SRC activation in MPH subjected to ERS using phosphorylation levels of SRC at Y416 (p-Y416-SRC) as a marker. We found that ERS leads to SRC activation in MPH, which could be prevented by treatment with its inhibitor PP2 (Fig 5H). Importantly, this was associated with a dampening of ERS-mediated degradation of HNF4A (1 h after ERS induction; Fig 5F and I, Appendix Fig S18B) and of subsequent (4 h after ERS induction) LIVER-ID TF gene repression (Fig 5J).

These data indicate that ERS triggers loss of hepatic identity through a global impairment of LIVER-ID TF expression/activities, involving EIF2AK3/PERK and SRC-dependent rapid decrease in their protein levels.

## Sustained loss of LIVER-ID genes and concomitant induction of ERS gene expression in dysfunctional mouse and human livers

To assess whether similar responses occur upon acute liver injury, we performed experiments on injured/dysfunctional livers of mice subjected to bacterial injection (sepsis[BIM] model; Paumelle *et al*, 2019). Our results show that sepsis triggers profound alterations in the liver transcriptome (Appendix Fig S19A) compatible with loss of hepatic molecular identity (Fig 6A compared to 1C). Indeed, sepsis also decreased expression of LIVER-ID genes (Fig 6B and Appendix Fig S19B–F), concomitant with an enrichment for ERS UP genes among the most strongly upregulated genes in septic mouse livers (Fig 6C). Similar observations were made when mining the transcriptomic response occurring in other mouse models of liver damage including drug-induced liver injury (Appendix Fig S20).

Overall, impaired LIVER-ID gene expression in injured livers was linked to partial hepatic dedifferentiation as judged using principal component analysis (Fig 6D). Indeed, the transcriptome of injured livers resembles more that of newborn livers than that of mature adult livers, in line with postnatal liver maturation being linked to significant transcriptomic changes (Fig 1N; Bhate *et al*, 2015; Peng *et al*, 2017).

Further mining the transcriptional changes induces by sepsis, we observed that genes significantly modulated by ERS in MPH (from Fig 1) showed a similar regulation pattern in liver (Fig 6E, Appendix Fig S21A and B) or purified hepatocytes (Appendix Fig S21C–E) from septic mice. Importantly, LIVER-ID TF expression was globally compromised (Fig 6F and G, Appendix Fig S21F and G), accompanied by a switch in the expression of the PAR-bZIP TF family members (Appendix Fig S22A) and reduced expression of genes involved in xenobiotic metabolism (Appendix Fig S22B). Pre-treatment of septic mice with the ERS inhibitor tauroursodeoxycholic acid (TUDCA) blunted ERS and displayed hepatoprotective effects, as indicated by reduced XBP1S (Fig 6H and I, Appendix Fig S22D) and decreased levels of circulating aminotransferases (Fig 6J), respectively. Interestingly, TUDCA concomitantly allowed to protect from a general loss of LIVER-ID TF expression (Fig 6H and I, Appendix Fig S22D and E), pointing to the functional link between ERS and loss of LIVER-ID gene expression in injured liver.

To monitor ERS and LIVER-ID gene expression upon liver recovery, we made use of a second sepsis mouse model (denoted sepsis[CLP]), which combines cecal ligation and puncture to induce sepsis with intravenous fluid resuscitation in order to mimic the clinical setting encountered in intensive care units (ICUs; Derde *et al*, 2017). Indeed, this model allows an assessment of transcriptional changes in both the acute phase and post-acute resolutive

phase (3 days after CLP) of liver injury (Thiessen *et al*, 2017). ERS gene upregulation and LIVER-ID TF repression, together with a switch in PAR-bZIP TF expression, were observed in the acute phase of sepsis (10 h after CLP; Fig 7A and Appendix Fig S22F). At later timepoints, transcriptional changes were blunted with most genes returning to near baseline levels 3 days after CLP (Fig 7A). Hence, reminiscent of liver regeneration following PHx, loss of molecular identity together with ERS gene induction transiently co-occur in the mouse liver upon sepsis. This indicated that ERS resolution was phased with recovery of the LIVER-ID program. To further assess the functional relationship between recovery from ERS and loss of liver-identity, we mined mouse liver transcriptomic data obtained from tunicamycin-injected WT or $Atf6^{-/-}$ (Atf6 KO) mice, the latter being characterized by an inability to resolve ERS (Arensdorf *et al*, 2013). Indeed, while ERS UP genes have returned to baseline levels

34 h after ERS in WT mice, their decreased expression is only partial in $Atf6^{-/-}$ mice leading to sustained ERS UP gene levels (Fig 7B). This was accompanied by a failure of ERS DOWN and LIVER-ID genes to fully return to baseline levels in $Atf6^{-/-}$ compared with WT mice (Fig 7C). This indicated that sustained ERS impedes re-establishment of the LIVER-ID program. To define whether sustained loss of LIVER-ID gene expression is linked to liver dysfunction, we mined transcriptomic data from hepatocyte-specific HNF4A KO mice subjected to PHx, since these mice fail to recover ultimately leading to their death (Fig 7D; Huck *et al*, 2019). Importantly, we found that this was linked to sustained downregulation of LIVER-ID genes and TFs 5 days after PHx (Fig 7D) at a time when the hepatic program is normally re-established (Fig 1A). These data therefore indicate that LIVER-ID TF re-expression is critically required to preclude detrimental consequences of liver injury.

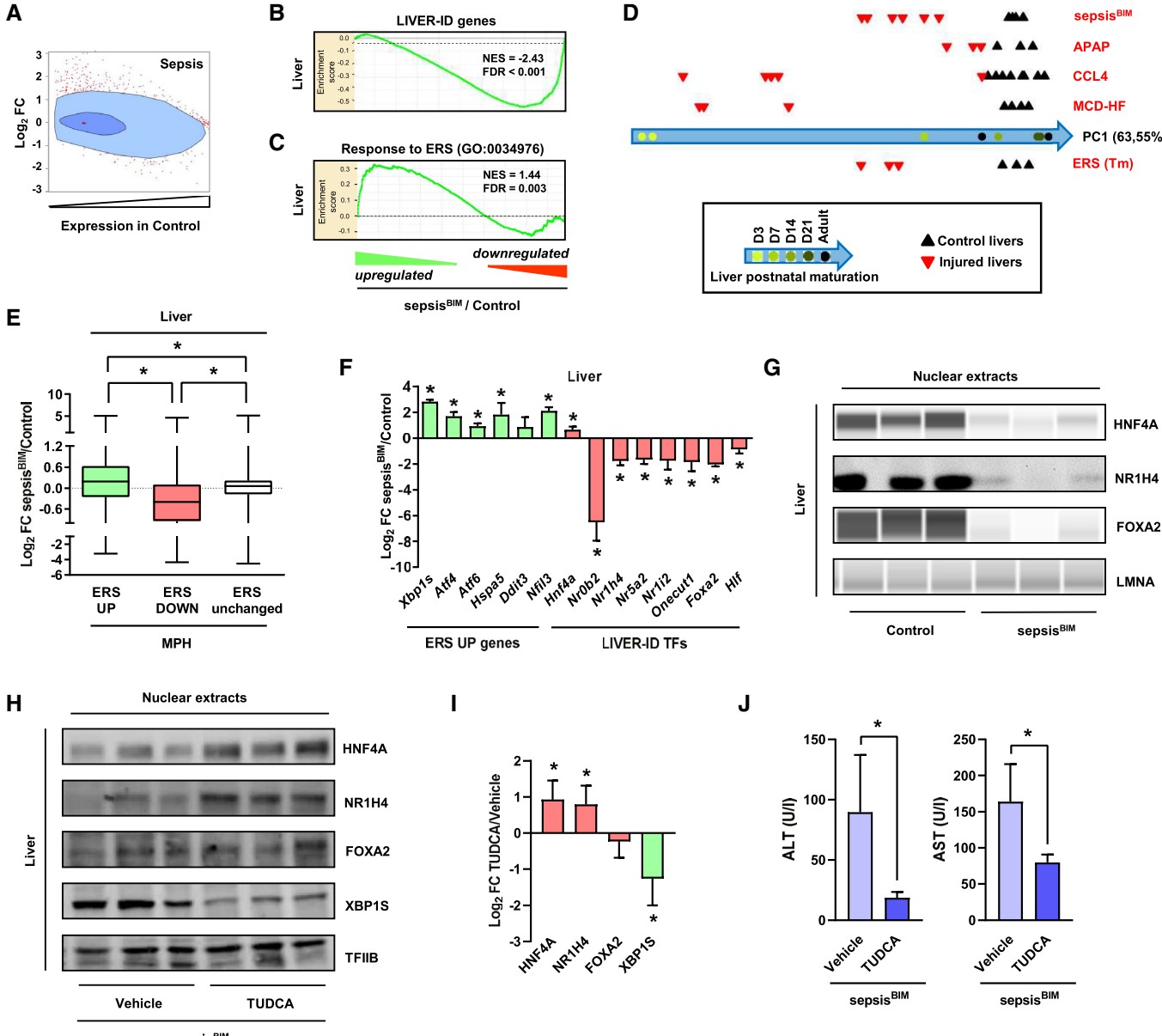

**Figure 6.**

**Figure 6. ERS contributes to LIVER-ID gene downregulation and loss of hepatic identity in septic mice.**

A   Similar analysis as in Fig 1C using transcriptomic data from the liver of sepsis[BIM] mice (16 h after intraperitoneal injection of live *E. coli*).

B, C   Enrichment plots from GSEA performed using LIVER-ID genes (B) or the response to ERS gene set (GO:0034976) (C) as the gene set and transcriptomic differences in sepsis[BIM] vs control mouse liver as the ranked gene list.

D   Comparison of the transcriptome of the indicated liver injury models (details are provided in Appendix Fig S20 and Table EV6) with that of the developing mouse liver performed as described in the Materials and Methods. PC1 is the first principal component which represents 63.55% of the variability within the mouse liver differentiation study. PC1 was used to project the liver injury studies. APAP, acetaminophen overdose—model of drug-induced acute liver injury. CCL4, carbon tetrachloride hepatotoxicity—model of drug-induced chronic liver injury. MCD-HF, methionine–choline-deficient diet with high fat—model of NASH/fibrosis.

E   Box plots showing Log$_2$ FC for ERS UP, DOWN, or unchanged genes in sepsis[BIM] vs control mouse liver (6 mice per group). Box plots are composed of a box from the 25$^{th}$ to the 75$^{th}$ percentile with the median as a line and min to max as whiskers. One-way ANOVA with Welch's correction and Dunnett's modified Tukey–Kramer pairwise multiple comparison test was used to assess statistical significance, *$P < 0.05$.

F   RT–qPCR analyses of selected ERS UP genes and LIVER-ID TFs monitoring expression changes in the liver of sepsis[BIM] vs control mice (6 mice per group). The bar graph shows means ± SD (standard deviations). One-sample *t*-test with BH correction for multiple testing was used to determine whether the mean Log$_2$ FC sepsis[BIM]/control is statistically different from 0, *$P < 0.05$.

G   Nuclear extracts from livers of control or sepsis[BIM] mice were subjected to Western blot or Simple Western immunoassay with antibodies against HNF4A, NR1H4, or FOXA2. LMNA was used as loading control.

H   Nuclear extracts from livers of sepsis[BIM] mice pre-treated for 4 consecutive days with vehicle or 500 mpk TUDCA were subjected to Western blot with antibodies against HNF4A, NR1H4, FOXA2, or XBP1S. TFIIB was used as loading control. Additional mice are shown in Appendix Fig S22D.

I   Densitometric quantification of the protein expression data from 6 mice per condition shown in panel (H) (3 mice per condition) and Appendix Fig S22D (3 additional independent mice per condition). The bar graph shows means ± SD (standard deviations). One-sample *t*-test with BH correction for multiple testing was used to determine whether the mean Log2 FC TUDCA/Vehicle is statistically different from 0, *$P < 0.05$.

J   Serum alanine aminotransferase (ALT) (*left*) and aspartate aminotransferase (AST) activities (*right*) from sepsis[BIM] mice pre-treated for 4 consecutive days with vehicle or 500 mpk TUDCA (10 mice per group). The bar graphs show means ± SD (standard deviations). Student's *t*-test was used to assess statistical significance, *$P < 0.05$.

Despite intense ICU care, septic patients frequently present with liver failure leading to their death (Nesseler *et al*, 2012). Interestingly, livers from deceased septic humans displayed downregulation of LIVER-ID TF encoding genes and concomitant upregulation of ERS UP genes when compared to control donors (undergoing elective restorative rectal surgery; Appendix Fig S23A and B). Moreover, correlative analyses revealed that LIVER-ID TFs behave as a group of genes with strong positive correlation, which are overall inversely correlated with ERS UP genes (including *NFIL3*), in livers of septic humans (Fig 7E). To further assess whether loss of hepatic

molecular identity might contribute to human liver dysfunction, we compared LIVER-ID TF gene expression levels between livers from deceased septic patients with serum bilirubin levels below (Bil < 2) or above (Bil > 2) 2 mg/dl [i.e., the most commonly used cut-off in clinics to define liver dysfunction in septic patients (Vincent *et al*, 1996)]. We observed a stronger overall decrease in LIVER-ID TF gene expression in the Bil > 2 group (Fig 7F), differences being the most pronounced for *NR1I2/PXR* and *HLF* (Fig 7G). This was associated with a more pronounced switch in the expression of the PAR bZIP TF family in the Bil > 2 group compared with the Bil < 2 group

**Figure 7. Competitive equilibrium between LIVER-ID and ERS gene expression in injured mouse and human livers.**

A   RT–qPCR analyses of selected ERS UP genes and LIVER-ID TFs in livers from sepsis[CLP] mice collected 10, 30 h, or 3 days after CLP (15 mice per group) vs livers from healthy pair-fed mice (control) (15 mice per group). The bar graph shows means ± SD (standard deviations). Wilcoxon test with BH correction for multiple testing was used to assess statistical significance, *$P < 0.05$.

B   Box plots showing Log$_2$ FC ERS/control in mouse liver for ERS UP genes 8 and 34 h after tunicamycin injection in WT or ATF6 KO mice (3 mice per experimental condition). Box plots are composed of a box from the 25$^{th}$ to the 75$^{th}$ percentile with the median as a line and min to max as whiskers. One-sample *t*-test with BH correction for multiple testing was used to determine whether the mean Log$_2$ FC ERS/control is statistically different from 0, *$P < 0.05$. NS, not significant.

C   Similar analyses to panel (B) for ERS DOWN and LIVER-ID genes (3 mice per experimental condition). Box plots are composed of a box from the 25$^{th}$ to the 75$^{th}$ percentile with the median as a line and min to max as whiskers. One-sample *t*-test with BH correction for multiple testing was used to determine whether the mean Log$_2$ FC ERS/control is statistically different from 0, *$P < 0.05$.

D   Main observations from Huck *et al* (2019) (*left*) and box plots showing Log$_2$ FC HNF4A KO/control in mouse liver (three mice per group) for LIVER-ID genes and TFs 5 days after PHx (*right*). Box plots are composed of a box from the 25$^{th}$ to the 75$^{th}$ percentile with the median as a line and min to max as whiskers. One-sample *t*-test with BH correction for multiple testing was used to determine whether the mean Log$_2$ FC HNF4A KO/control is statistically different from 0, *$P < 0.05$.

E   Correlations of gene expressions within the critically ill group were used to organize the analyzed genes as a network. Green bars indicate a positive correlation, while red bars indicate a negative correlation. The color intensity is proportional to the correlation coefficient. The position of the genes is determined by both the directions and values of the correlation coefficients.

F   Expression of LIVER-ID TF encoding genes from Appendix Fig S23A was analyzed in the livers of deceased critically ill patients with sepsis displaying agonal bilirubin levels below (Bil < 2; *n* = 34) or above (Bil > 2; *n* = 28) 2 mg/dl. Median fold change expression level of each group has been used to generate the box plots. Expression levels in the critically ill groups are expressed relative to those in the control group (*n* = 18). Box plots are composed of a box from the 25$^{th}$ to the 75$^{th}$ percentile with the median as a line and min to max as whiskers. One-tailed *t*-test was used to assess whether expression of LIVER-ID TF encoding genes in the Bil > 2 group is significantly greater than in the Bil < 2 group, *$P < 0.05$.

G   RT–qPCR analyses of indicated LIVER-ID TF encoding genes monitoring expression in the livers of Bil < 2 (*n* = 34) or Bil > 2 (*n* = 28) groups of deceased critically ill patients with sepsis *vs* control donors (*n* = 18). Data are shown as box plots, with mRNA levels of the critically ill groups expressed relative to those of the control group. Box plots are composed of a box from the 25$^{th}$ to the 75$^{th}$ percentile with the median as a line and min to max as whiskers. Wilcoxon test was used to assess statistically significant differences with the Bil > 2 group, *$P < 0.05$.

H   RT–qPCR analyses of indicated ERS UP genes performed and analyzed as in panel (G). Box plots are composed of a box from the 25$^{th}$ to the 75$^{th}$ percentile with the median as a line and min to max as whiskers. Wilcoxon test was used to assess statistically significant differences with the Bil > 2 group, *$P < 0.05$.

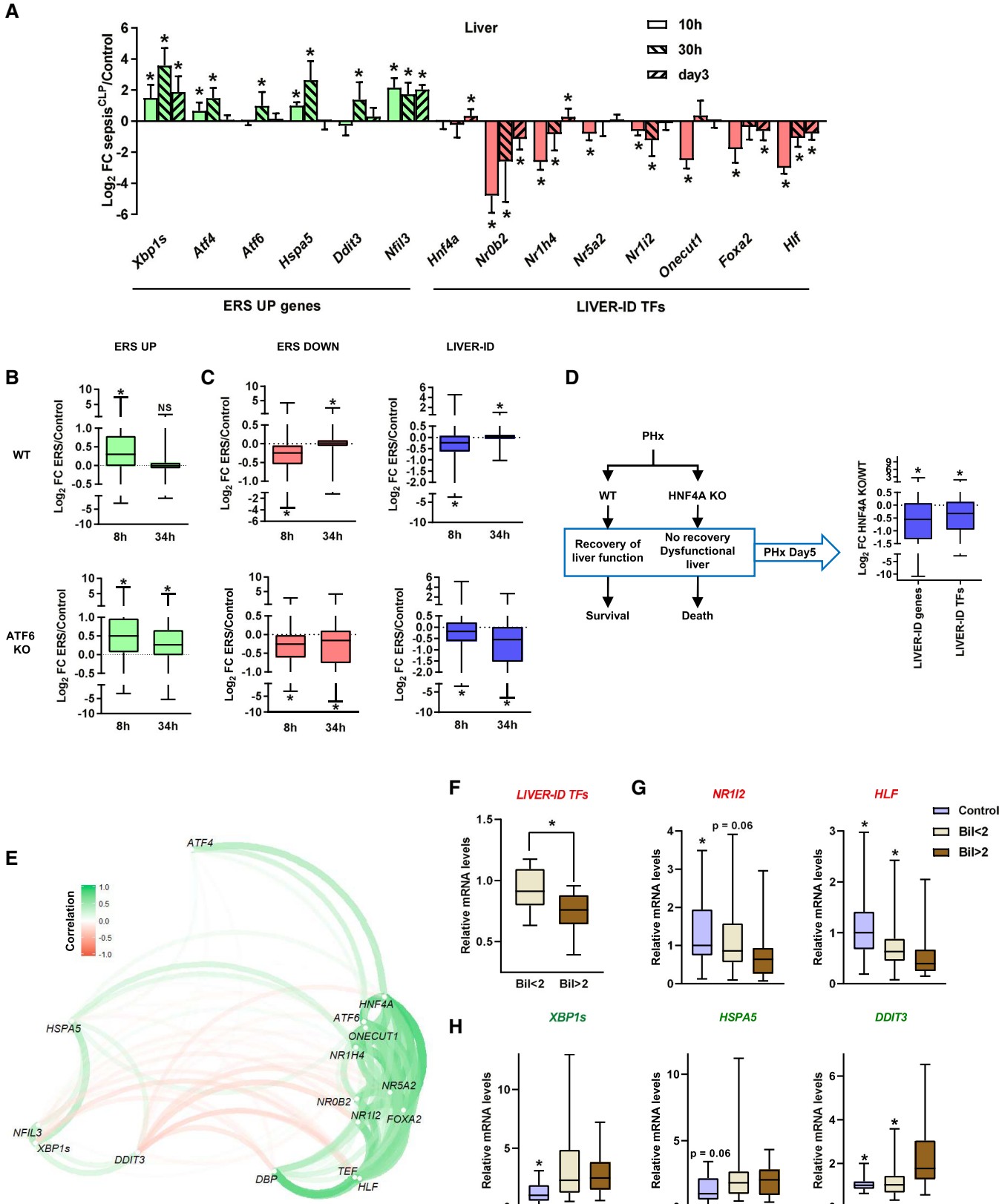

Figure 7.

(Appendix Fig S23C). While ERS gene induction was present in the two groups, *DDIT3/CHOP* was only upregulated in the Bil > 2 group, suggestive of a more severe ERS and/or of activation of additional detrimental signaling pathways which would add up to ERS in patients with liver dysfunction (Fig 7H).

Altogether, these data indicate that sustained loss of LIVER-ID TF expression linked to persistent ERS gene induction is detrimental to liver function recovery, which may relate to liver dysfunction in septic patients.

## Discussion

ERS had previously been shown to repress a handful of genes involved in liver metabolic functions (Chikka *et al*, 2013). Here, we have redefined the paradigm related to acute ERS-induced changes in the liver by pointing to a more global loss of molecular identity

and partial hepatic dedifferentiation, which we found to be characteristic of acute liver injury. As discussed hereafter and detailed in Fig 8A, loss of LIVER-ID gene expression results from several ERS-induced signaling pathways, consistent with signaling from the different sensors of ERS being activated in liver injury (Wang *et al*, 2020) and functionally intermingled (Fig 5 and Appendix Fig S24; Brewer, 2014). Importantly, the relevance of our findings is indicated by several lines of evidence defining chemically induced ERS as appropriate for the study of pathophysiological ERS-induced transcriptional regulations. First, repression of LIVER-ID genes was observed both with tunicamycin and with thapsigargin, ruling out any drug-specific artifact (e.g., Appendix Fig S3G). Second, this repression was readily linked to ERS and not to any other potential drug-related effect as it was blunted by (i) cycloheximide (Appendix Fig S25A), which, by inhibiting protein synthesis, alleviates ERS and decreases UPR gene expression (Harding *et al*, 2000); (ii) the chemical chaperone PBA (Fig 2G); and (iii) inhibitors of the

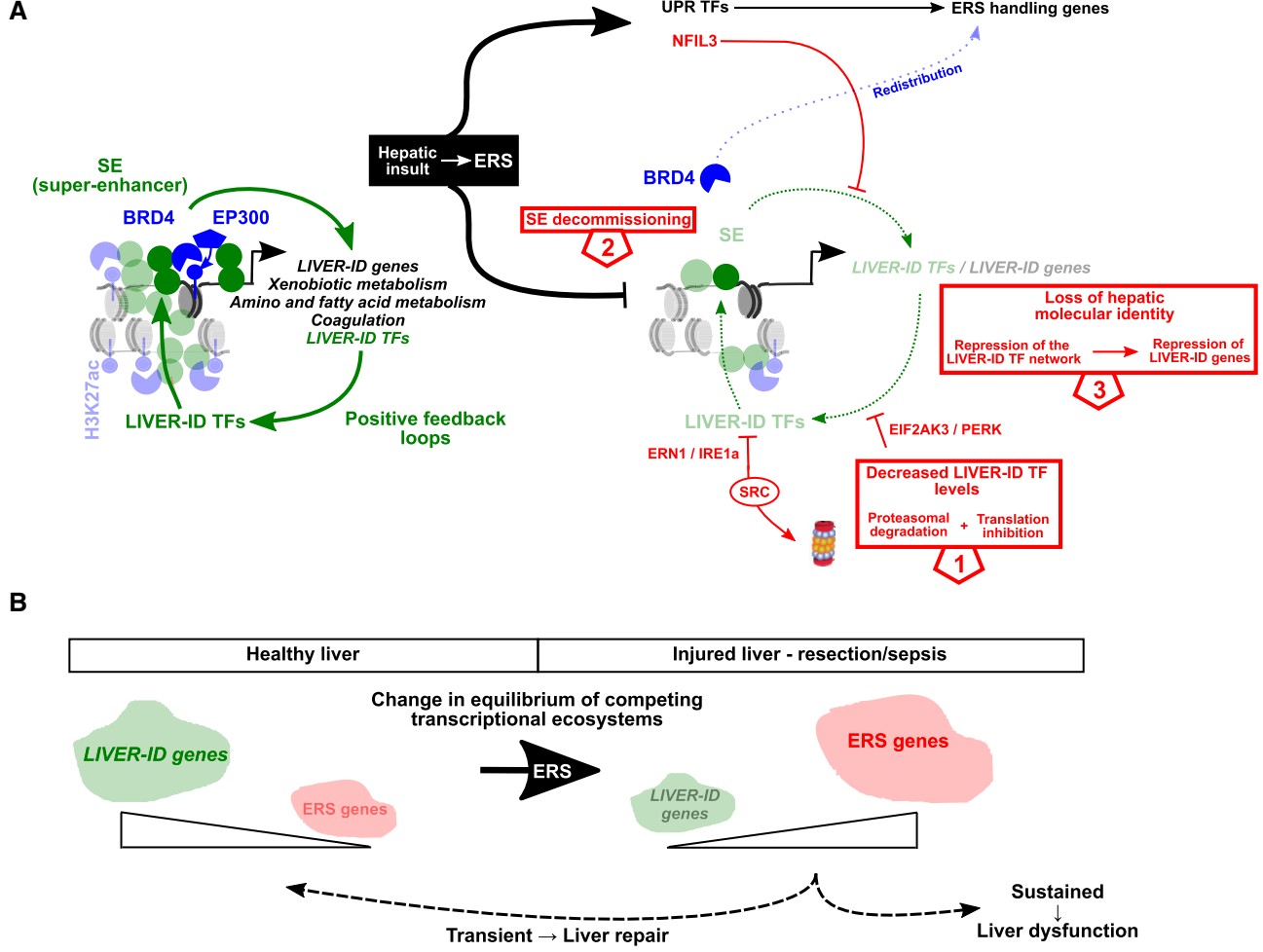

**Figure 8. Summary of the molecular mechanisms involved in ERS-induced loss of hepatic molecular identity and implications for liver pathophysiology.**

A  Model for ERS-mediated loss of hepatic identity. Initial loss in LIVER-ID TF activities involves reduced protein levels linked to PERK and SRC signaling ❶, which is secondarily amplified by decommissioning of BRD4 SE ❷ and impaired feedback loops within the LIVER-ID TF hepatic network ❸. This consequently leads to loss of hepatic molecular identity and partial dedifferentiation of hepatocytes ❸. Induction of NFIL3 operates as an additional mechanism further contributing to active repression of the LIVER-ID TF network target genes, especially those involved in xenobiotic metabolism.

B  Implications of the competitive equilibrium between the hepatic and ERS transcriptional programs in liver pathophysiology. Please also refer to the discussion.

different ERS sensors (Fig 5D and E, Appendix Fig S24). Third, chemically induced ERS recapitulated both induction of the different arms of the UPR (Figs 1L and M, and 6F) and the overall preferential downregulation of LIVER-ID genes observed in injured liver (Figs 1 and 6, Appendix Fig S19C–F). Fourth, this repression of LIVER-ID genes was not artificially linked to strong chemically induced ERS. Indeed, dose–response experiments showed that ERS-mediated repression was proportional to the ERS intensity, i.e., not requiring maximal ERS response (Appendix Fig S25B). Moreover, induction of *Klf9*, recently described as a marker of strong ERS (Fink *et al*, 2018), was not stronger in the chemically induced ERS models (Appendix Fig S25C). Fifth, inhibition of SRC kinase prevents HNF4A degradation both following acute ERS (Fig 5F and I, Appendix Fig S18) and liver PHx (Huck *et al*, 2019).

In addition to translation inhibition induced by EIF2AK3/PERK, our study points to initial loss in LIVER-ID TF expression involving ERS-mediated proteasomal degradation events. In particular, we report a role for SRC kinase-mediated proteasomal degradation of HNF4A. These findings, which are in contrast with a previous study suggesting that HNF4A protein levels were not modulated by ERS (Arensdorf *et al*, 2013), are of importance when taking into account HNF4A's requirement for maintaining/establishing the LIVER-ID TF network and hepatocyte identity (Fig 7D; Lau *et al*, 2018; Huck *et al*, 2019; Thakur *et al*, 2019). Additional mechanisms directing LIVER-ID TFs toward degradation most probably co-occur such as those triggered by interaction with XBP1S (Zhou *et al*, 2011). More generally, our findings are however in line with a recent study indicating that targets of endoplasmic reticulum-associated protein degradation also include TFs and epigenetic regulators (Wei *et al*, 2018). Loss of LIVER-ID TF expression is further amplified by their transcriptional downregulation being linked to their organization as an interdependent transcriptional network where LIVER-ID TF encoding genes form auto- and cross-regulatory loops (Appendix Fig S16; Fig 8A). The hepatic TF network increases in complexity and stability during development making LIVER-ID TF gene expression more robust to alterations in single TF expression/activity in the adult liver (Kyrmizi *et al*, 2006). Our finding that deletion of HNF4A dramatically impacts re-expression of the other LIVER-ID TFs involved in termination of liver regeneration following PHx (Fig 7D) indicates that liver stress/injury decreases the hepatic core TF network robustness. This is most probably linked to the global loss of LIVER-ID TF expression jeopardizing cross-regulatory loops therefore rendering hepatocytes more sensitive to loss of activities of individual TFs (Gjuvsland *et al*, 2007; Felix & Barkoulas, 2015). ERS-induced transcriptional downregulation of LIVER-ID TFs/genes involves decommissioning of BRD4 at LIVER-ID TF densely co-bound CRMs organized into SE. Importantly, this implies that not all LIVER-ID genes are equally sensitive to ERS-mediated repression defining partial hepatic dedifferentiation as being linked to preferential repression of highly expressed SE-associated LIVER-ID genes. TF recruitment complexity and SE have both been positively associated with particularly strong activity of regulatory regions important for cell identity (Whyte *et al*, 2013; Santiago-Algarra *et al*, 2017). In this context, we show that a widespread decrease in LIVER-ID TFs accounts for the breadth of detrimental transcriptional effects of ERS/liver injury on hepatic molecular identity.

Contrary to a recent study suggesting that BRD4 is required for establishment but not maintenance of cell identity (Lee *et al*, 2017),

and in line with other previous reports (Di Micco *et al*, 2014), our work indicates a role for BRD4 in maintenance of mature hepatocyte molecular identity. ERS-induced loss of hepatic molecular identity is accompanied by BRD4 redistribution toward ERS gene regulatory regions. Competition for transcriptional resources has been proposed to rule transient transcriptional adaptations to environmental disturbances in a model referred to as transcriptional ecosystem (Silveira & Bilodeau, 2018). In line, we have found that stressed/injured liver requires to enhance expression of an exceptionally large spectrum of genes, which may trigger hepatocytes to temporarily decommission highly active regulatory regions to supply the required transcriptional resources (Fig 8A and B). Squelching (competition between TFs for limited cofactor amounts) has been proposed, but never firmly proven, to be a driving force in cofactor redistribution responsible for trans-repression (Schmidt *et al*, 2016). This was for instance suggested as an explanation for transcriptional repression induced by tumor necrosis factor (Schmidt *et al*, 2015). In the context of acute ERS, our study points to a cascade of molecular events described here above involving early loss of LIVER-ID TF expression/activities, indicating repression of LIVER-ID gene expression does not rely *per se* on squelching, i.e., ERS TFs competing off BRD4 binding from fully active LIVER-ID TFs. Together with the role of NFIL3, our data rather indicate that ERS-mediated repression is an active process and not an indirect consequence of ERS gene induction. While squelching on its own cannot explain loss of LIVER-ID gene expression, our data do not entirely rule out that competition for BRD4 might further contribute to LIVER-ID gene downregulation.

Our data using several models of liver injury and genetically deficient mice have established that the hepatic and ERS transcriptional programs are in competitive equilibrium with direct relevance toward the liver's ability to recover from injury. Indeed, while transient loss of hepatic molecular identity is linked to stress/injury handling, detrimental effects occur if the hepatic transcriptional program cannot be re-established. In line with loss of LIVER-ID TF expression being instrumental in triggering liver injury, forced hepatic expression of HNF4A or FOXA2 has been shown to be protective (Wang *et al*, 2017; Huck *et al*, 2019). Predictive models of transcriptomic regulation where alternative programs are in competitive equilibrium postulate that cessation of environmental disturbances would allow cells to restore their normal transcriptional program as the default one (Silveira & Bilodeau, 2018; Fig 8B). The kinetic and feedback mechanisms allowing re-establishment of the hepatic identity upon recovery from liver injury remain to be fully defined but critically require LIVER-ID TFs re-expression (Fig 7A–D). Interestingly, ERS-induced loss of molecular identity is not restricted to the liver (Appendix Fig S26 and Table EV4), but whether this mechanism is globally involved in organ repair and/or dysfunction beyond liver remains to be defined. The mechanisms leading to organ dysfunction in sepsis are still incompletely understood. Cell death has been ruled out as a potential main contributor (Takasu *et al*, 2013). In this context, our results obtained from the livers of deceased septic patients point to sustained loss of LIVER-ID TFs as a likely contributor to human liver dysfunction (Fig 8B). Switch in the expression of the PAR-bZIP family members may relate to compromised xenobiotic metabolism associated with poor outcome in treated critically ill septic patients (Gachon *et al*, 2006; Woznica *et al*, 2018) while loss of PPARA may impede liver's ability to adapt its metabolic activities during sepsis

(Paumelle *et al*, 2019). While our data indicate that acute ERS directly represses LIVER-ID gene expression (events occurring in MPH in the absence of detectable activation of regulatory regions involved in the hepatic inflammatory response (Appendix Fig S27)), additional signals including inflammatory cytokines (Brown *et al*, 2014) most probably combine with ERS during sepsis to trigger the exceptionally large alterations to the hepatic transcriptional equilibrium we observed in this condition.

# Materials and methods

### Reagents and Tools table

| Reagent/Resource | Reference or source | Identifier or catalog number |
|---|---|---|
| **Experimental models** | | |
| C57BL/6J (*M. musculus*) | Charles River | C57BL/6J |
| C57BL/6J (*M. musculus*) Nfil3$^{-/-}$ | van der Kallen *et al* (2015) | N/A |
| AML12 cells (*M. musculus*) | ATCC | CRL-2254 |
| HEK293 (*H. sapiens*) | ATCC | CRL-1573 |
| **Recombinant DNA** | | |
| pcDNA3.1-mNFIL3 | Addgene | Cat # 34572 |
| pSGG5-hHNF4A | Suaud *et al* (1999) | N/A |
| **Antibodies** | | |
| *example*: Rabbit-anti-H3 | Abcam | Cat # ab1791 |
| ACTB | Sigma-Aldrich | Cat # A5441 (WB) |
| BRD4 | Bethyl Labs | Cat # A301-985A100 (ChIP) |
| BRD4 | Wu *et al* (2006) | N/A (WB) |
| DDIT3 | Santa Cruz | Cat # sc7351 (WB) |
| FOXA2 | Abcam | Cat # ab23630 (WB, WES) |
| Histone H3 | Cell Signaling | CST 3638S (WB) |
| H3K27ac | Active Motif | Cat # 39685 (ChIP) |
| HNF4A | Perseus | Cat # PP-H1415-00 (WB, WES) |
| LMNA | Santa Cruz | Cat # sc20681 (WB, WES) |
| NFIL3 | Cell Signaling | Cat # CST 14312S (WB) |
| NR1H4 | Perseus | Cat # PP-A9033A-00 (WB) |
| p-Y416-SRC | Cell Signaling | Cat # CST 6943S (WB) |
| SRC | Cell Signaling | Cat # CST 2109S (WB) |
| XBP1s | Cell Signaling | Cat # CST 12782S (WES) |
| TFIIB | Santa Cruz | Cat # sc7225 (WB) |
| EP300 | Active motif | Cat #61401 (IP) |
| EP300 | Santa Cruz | Cat # sc-585 (WES) |
| IgG | Santa Cruz | Cat # sc-2025 (IP) |
| HRP-conjugated anti-mouse | Sigma-Aldrich | Cat # A4416 (WB) |
| HRP-conjugated anti-rabbit | Sigma-Aldrich | Cat # A0545 (WB) |
| 12–230 kDa Wes separation module | ProteinSimple | Cat# SM-W004 |
| 66–440 kDa Jess or Wes Separation Module | ProteinSimple | Cat # SM-W008 |
| **Oligonucleotides and sequence-based reagents** | | |
| qPCR primers | This study | Table EV5 |
| **Chemicals, enzymes and other reagents** | | |
| *example*: T7 Endonuclease I | New England Biolabs | Cat # M03020S |
| Thapsigargin | Sigma-Aldrich | Cat # T9033 |

**Table** (continued)

| Reagent/Resource | Reference or source | Identifier or catalog number |
|---|---|---|
| Tunicamycin | Sigma-Aldrich | Cat # T7765 |
| MG132 | Sigma-Aldrich | Cat # M7449 |
| PP2 | Sigma-Aldrich | Cat # P0042 |
| Cycloheximide | Sigma-Aldrich | Cat # C7698 |
| JQ1 | Sigma-Aldrich | Cat # SML1524 |
| Trichostatin A | Sigma-Aldrich | Cat # T8552 |
| C646 | Sigma-Aldrich | Cat # SML0002 |
| MZ1 | Tocris | Cat # 6154 |
| TUDCA | Sigma-Aldrich | Cat # T0266 |
| PBA | Sigma-Aldrich | Cat # SML0309 |
| STF083010 | Sigma-Aldrich | Cat # SML0409 |
| ISRIB | Sigma-Aldrich | Cat # SML0843 |
| AEBSF | Euromedex | Cat # 50985 |
| jetPEI | Polyplus Transfection | Cat # 101-40 |
| High capacity cDNA reverse transcription kit | Applied Biosystems | Cat # 4368813 |
| Brilliant II Sybr Green QPCR MAster mix | Agilent Biotechnologies | Cat # 600831 |
| Disuccinimidyl glutarate (DSG) | Thermo Fischer Scientific | Cat # 20593 |
| Formaldehyde | Sigma-Aldrich | Cat# F8775 |
| Benzonase | Sigma-Aldrich | Cat# E1014 |
| Protein A Sepharose | GE Healthcare | Cat # GE17-1279-01 |
| Protein G Sepharose | GE Healthcare | Cat # GE17-0618-01 |
| Complete protease inhibitor cocktail | Roche | Cat # 11836145001 |
| Phosphatase inhibitor cocktail | Sigma-Aldrich | Cat # P044 |
| Proteinase K | Qiagen | Cat # 19133 |
| tRNA | Sigma-Aldrich | Cat # R5636 |
| William's medium (MPH) | Lonza | Cat# BE12761F |
| DMEM/F12 (AML12) | Dutcher | Cat # P04-41250 |
| DMEM (HEK293) | Gibco-Life Technologies | Cat #31966 |
| Collagen | Roche | Cat# 11179179001 |
| Collagenase | Sigma-Aldrich | Cat # C5138 |
| Yeast tRNA | Sigma-Aldrich | Cat # R5636 |
| Extract-All (Trizol) | Eurobio | GEXEXT04-0U |
| **Software** | | |
| Partek Genomics Suite 6.6 | Partek | N/A |
| DESeq 1.26.0 | Anders and Huber (2010) | N/A |
| FactoMineR 1.41 | Lê *et al* (2008) | N/A |
| STEM v1.3.11 | Ernst *et al* (2005) | N/A |
| GIANT | Vandel *et al* (2018) | N/A |
| Galaxy | Afgan *et al* (2018) | N/A |
| R | R Core Team (2015) | N/A |
| aplpack v1.3.2 R package | R Core Team (2015) | N/A |
| ToppGene Suite | Chen *et al* (2009) | N/A |
| GSEA v3.0 | Subramanian *et al* (2005) | N/A |
| BubbleGUM v1.3.19 | Spinelli *et al* (2015) | N/A |
| Bowtie 2 | Langmead and Salzberg (2012) | N/A |

    

**Table** (continued)

| Reagent/Resource | Reference or source | Identifier or catalog number |
|---|---|---|
| Integrated Genome Browser (IGB 9.0.1) | Freese *et al* (2016) | N/A |
| MACS2 | Chen *et al* (2015) | N/A |
| Rank Ordering of Super-Enhancers (ROSE) | Loven *et al* (2013), Whyte *et al* (2013) | N/A |
| csaw v1.6.1 | Lun and Smyth (2014, 2016) | N/A |
| Locus Overlap Analysis (LOLA 1.4.0) | Sheffield and Bock (2016) | N/A |
| gplots v3.0.1 | Warnes *et al* (2016) | N/A |
| graphics R package | R Core Team (2015) | N/A |
| Prism v5 and 8 | GraphPad | N/A |
| GeneSnap v7.12.06 | Syngene | N/A |
| Image Studio Lite v5.2 | LI-COR Biosciences | N/A |
| Compass | ProteinSimple | N/A |
| bioDBnet | Mudunuri *et al* (2009) | N/A |
| **Other** | | |
| Influx sorter | Becton Dickinson | N/A |
| Agilent 2100 Bioanalyzer | Agilent Biotechnologies | N/A |
| MoGene-2_0-st | Affymetrix | N/A |
| GeneChip™ Scanner 3000 7G | Applied Biosystems | Cat# 00-0210 |
| GeneChip™ Fluidics Station 450 | Applied Biosystems | Cat# 00-0079 |
| Bioruptor Pico | Diagenode | Cat# B01060010 |
| MinElute PCR purification kit | Qiagen | Cat# 2800 |
| ALT activity | Thermo Fischer Scientific | Cat# 981769 |
| AST activity | Thermo Fischer Scientific | Cat# 981771 |
| KONELAB 20 | Thermo Fischer Scientific | Cat# 981801 |
| Illumina Hi-seq 4000 | Illumina | N/A |
| G-box | Syngene | N/A |
| Simple Western, WES system | ProteinSimple | N/A |
| TnT® Quick Coupled Transcription/Translation System | Promega | Cat# L5020 |
| iBright™ CL1500 Imaging System | Thermo Fischer Scientific | A44240 |

## Methods and Protocols

### Cell culture

The immortalized mouse hepatocyte cell-line AML12 was obtained from ATCC (CRL-2254) and cultured as previously described (Ploton *et al*, 2018). Mouse primary hepatocytes (MPH) were prepared from livers of 10-week-old male C57BL/6J mice (Charles River) as described in Bantubungi *et al* (2014) and grown on collagen-coated plates in serum-free William's medium (Ploton *et al*, 2018). Non-parenchymal cells (NPC) from the same livers were obtained by differential centrifugation. Briefly, liver homogenates obtained after perfusion were pressed through a 70-μm cell strainer and centrifugated for 5 min at 27 *g*. Pellets from this first centrifugation were washed and centrifuged twice again for 5 min at 27 *g* to obtain the MPH fraction. Supernatants from the first centrifugation were collected and centrifuged for 5 min at 400 *g* to obtain the NPC fraction. Separation of MPH and NPC was confirmed by monitoring expression of selected marker genes (Appendix Fig S2D). Acute endoplasmic reticulum stress (ERS) treatment in MPH is defined as 4-h treatment with 1 μM thapsigargin. Vehicle (0.04% DMSO) was used as control. In all figures from this study, ERS in MPH is defined as 4-h treatment with 1 μM thapsigargin unless indicated otherwise (shorter or longer treatment times with different concentrations were also used in some experiments as specifically indicated in the figures and their legends). Experiments involving MZ1 or JQ1 were performed by pre-treating MPH for 3 h with 0.01, 0.1, or 1 μM MZ1 or 1 h with 500 nM JQ1 before addition of 1 μM thapsigargin for 4 h. Experiments involving C646 were performed by co-treating MPH with 5, 10, or 20 μM C646 and 1 μM thapsigargin for 4 h. Experiments involving trichostatin A were performed by co-treating MPH with 1 μM trichostatin A and 1 μM thapsigargin for 4 h. Experiments involving cycloheximide were performed by co-treating MPH with 0, 1, or 10 μg/ml cycloheximide and 1 μM thapsigargin for 4 h. Experiments involving PBA, ISRIB, MG132, or PP2 were performed by pre-treating MPH for 30 min with 5 mM PBA, 1 μM ISRIB, 10 μM MG132, or 10 μM PP2 before addition of 1 μM thapsigargin for 1 or 4 h. Experiments involving inhibitors of the three arms of the UPR were performed

by pre-exposing AML12 cells to 30 μM STF083010, 200 μM ISRIB, or 100 μM AEBSF for 2 h and subsequently treating them for 4 h with 1 μM thapsigargin or 2 μg/ml tunicamycin.

Fluorescence-activated cell sorting (FACS)-isolated hepatocytes were obtained by directly running the MPH fraction into an Influx sorter (Becton Dickinson) equipped with a 200 μm nozzle and tuned at a pressure of 3.6 psi and a frequency of 6.3 kHz. Sample fluid pressure was adjusted to reach an event rate of 2,000 events/s. Hepatocytes were identified as FSChi SSChi events and sorted on a "pure" mode with 80% sorting efficiency.

HEK293 cells were grown as in Ploton *et al* (2018) and transfected using jetPEI (Polyplus Transfection) according to the manufacturer's instructions.

## Chemicals

All chemicals used in this study are provided in the Reagents and Tools table.

## Animal experiments

Male C57BL/6J wild-type (WT) mice were purchased from Charles River at 8 weeks of age and housed in standard cages in a temperature-controlled room (22–24°C) with a 12-h dark–light cycle. They had *ad libitum* access to tap water and standard chow and were allowed to acclimate for 2 weeks prior to initiation of the experimental protocol. ERS was induced by intraperitoneal injection of tunicamycin using 1 μg/g mouse body weight (Sigma-Aldrich, #T7765) or vehicle (150 mM dextrose), and liver was collected 8 h later (five mice per group). The *Nfil3*$^{-/-}$ (NFIL3 KO) mice used in this study (C57BL/6J background) were previously described (van der Kallen *et al*, 2015). WT littermates were used as controls. Mice of 10 weeks of age were treated with tunicamycin or vehicle as described above, and liver was collected 8 h after injection (eight mice per group). To induce ERS in muscle, 30 μg tunicamycin was injected intramuscularly into the gastrocnemius muscle. The contralateral leg was injected with a saline solution and used as control. Muscles were collected 24 h after injection (nine mice per group).

Two different models of sepsis were used. For the bacterial injection model (BIM) of sepsis (sepsis[BIM]), mice were injected intraperitoneally with $8 \times 10^8$ CFU of live *E. coli* (DH5α) bacteria or PBS (controls) and liver was collected 16 h later (six mice per group). In a separate experiment, mice were pre-treated for 4 consecutive days with tauroursodeoxycholic acid (TUDCA; intraperitoneal injection of 500 mpk/day) or vehicle (PBS) followed by bacterial injection 2 h after the last TUDCA administration on the fourth day (10 mice per group), and sacrificed 6 h after bacterial injection which is sufficient to induce LIVER-ID TF loss (Appendix Fig S22C). For the cecal ligation and puncture (CLP) model of sepsis (sepsis[CLP]), male C57BL/6J wild-type mice of 24 weeks of age were randomly allocated to sepsis[CLP] or healthy pair-fed control and sacrificed after 10, 30 h, or 3 days (15 mice per group per timepoint). Mice in the sepsis[CLP] groups were subjected to single-puncture CLP followed by intravenous fluid resuscitation as previously described (Derde *et al*, 2017). Briefly, mice were anesthetized, a catheter was inserted in the central jugular vein, and the surgical CLP procedure was performed (50% ligation of the cecum at half the distance between the distal pole and the base of the cecum and a single-puncture through-and-through) followed by intravenous fluid resuscitation. They received pain medication and antibiotics 6 h after CLP and from then on every 12 h for the remainder of the experiment and mice of the "day 3" group (prolonged phase) received parenteral nutrition from the morning after surgery to mimic the human clinical situation. The data reported for the sepsis[CLP] design correspond to the 10-h timepoint (acute phase) unless indicated otherwise. Healthy pair-fed mice were used as control.

All animal studies were performed in compliance with EU specifications regarding the use of laboratory animals and approved by the Nord-Pas de Calais Ethical Committee (for ERS treatments and the sepsis[BIM] design) or the KU Leuven Ethical Committee (P093/2014) (for the sepsis[CLP] design).

## Biochemical analyses

Plasma aspartate aminotransferase (AST) and alanine aminotransferase (ALT) activities were determined by colorimetric assays (Thermo Fischer Scientific) using serum obtained following retro-orbital blood collection.

## Real-time-quantitative PCR analyses of gene expression

RNA extraction, reverse transcription, and real-time-quantitative PCR (RT–qPCR) were performed as previously described (Dubois-Chevalier *et al*, 2017). The primer sequences are listed in Table EV5. All primers were designed to hybridize to different exons, and generation of single correct amplicons was checked by melting curve dissociation. Murine gene expression levels were normalized using hypoxanthine-guanine phosphoribosyltransferase (*Hprt*) (sepsis[CLP] experiments) or cyclophilin A (*PPia*) (all other experiments) housekeeping gene expression levels as internal control. Human gene expression levels were normalized using 18S ribosomal RNA (*RNA18S5*). For gene expression analyses, *in vitro* experiments (AML12 and MPH) were repeated at least three times (independent experiments), each experiment being performed in technical triplicates. For *in vivo* mouse studies, we used at least five animals per experimental condition (genotype or treatment). The number of biological replicates is indicated in the figure legends.

## Gene expression microarrays

RNA was extracted from MPH treated for 4 h with 1 μM thapsigargin (three independent experiments), livers of NFIL3 KO and WT littermates treated for 8 h with 1 μg/g tunicamycin (five mice per genotype per treatment), or gastrocnemius muscles of WT mice treated for 24 h with 30 μg tunicamycin (treated and contralateral control muscles from nine mice) and was checked for quantity and quality using the Agilent 2100 Bioanalyzer (Agilent Biotechnologies) before being processed for analysis using MoGene-2_0-st Affymetrix arrays according to the manufacturer's instructions. Data were analyzed as described hereafter and have been submitted to GEO under accession number GSE122508.

## Transcriptomic data analyses
### Liver-specificity index

The liver-specificity index was calculated as the difference in normalized expression in liver and mean of normalized expression in control tissues using data from BioGPS (Table EV6).

## Normalization of microarrays and identification of differentially expressed genes

Raw transcriptomic data from Affymetrix microarrays were normalized with Partek Genomics Suite 6.6 using background correction by

Robust Multi-array Average (RMA), quantile normalization, and summarization via median polish. Principal component analyses (PCA) were used for quality control of the data. RMA values were also used to display expression changes for selected gene sets in different figures. Differential expression analyses were performed at probeset level with Partek Genomics Suite. Dysregulated genes were defined taking into account any potential factor interaction in the original experimental design and using a Benjamini–Hochberg corrected *P*-value cut-off (FDR) set at 0.05.

### Single-cell RNA-seq data analyses

Raw counts from single-cell transcriptomic data (447 cells from E10.5 to E17.5; Yang *et al*, 2017) were normalized by estimation of library size factor with DESeq 1.26.0 (Anders & Huber, 2010) according to Brennecke *et al* (2013). PCA was performed on normalized data using FactoMineR 1.41 (Lê *et al*, 2008). Then, the average expression of ERS DOWN or ERS UP genes was projected for each cell on 2D PCA plot.

### Identification of preferential patterns of gene expression following partial hepatectomy

Genes with different temporal expression profiles were identified using the Short Time-series Expression Miner (STEM v1.3.11; Ernst *et al*, 2005), which fits dynamic patterns of gene expression to model profiles. Normalized gene expressions (rpkm) were obtained from Rib *et al* (2018), and average expression from replicates was used. Parameters were set at "log normalize data", 4 for "max unit change in model profiles between timepoints", $-0.05$ for "minimum absolute expression change", and FDR for "Correction method".

### Comparison of the breadth of transcriptomic changes occurring in the mouse liver

To make fold changes comparable with those obtained using RNA-seq, microarray data were normalized using the Affymetrix Power Tool (Thermo Fisher Scientific) run through the GIANT tools suite (Vandel *et al*, 2018) on a local instance of Galaxy (Afgan *et al*, 2018). Normalization was set to "scale intensity + rma" and normalization level to "probeset". Normalized expression values retrieved for the studies which used RNA-seq (Table EV6) were $log_2$-transformed. For each dataset, a single expression value per gene was defined using gene symbols as identifiers and by averaging values obtained from replicates. Fold changes ($log_2$) were next computed on scaled data, which were obtained using the scale function of the «graphics» R package (R Core Team, 2015) on each dataset separately. This was performed using global mean (mean of all expression values under all conditions of interest in a given study) for "center" parameter and global standard deviation for the "scale" parameter. Only genes common to all analyzed datasets were considered for subsequent analyses, and for each dataset, the bottom 20% genes with lowest expression in the liver were discarded. Bagplots were drawn using the "bagplot" function of the "aplpack" (v1.3.2) R package using default parameters (R Core Team, 2015). Bagplots are bivariate boxplots showing the spread of the data using a "bag" containing 50% of the data points with the largest depth (around the median) and its extension by a "loop" whose limit excludes outliers (Rousseeuw *et al*, 1999).

### Comparison of the transcriptome of injured livers with that of the developing mouse liver

Transcriptomic data of liver injuries were pooled, and batch effects were corrected with the "ComBat" function of the "sva" R pakage (Leek *et al*, 2019) using the mouse liver differentiation study as the batch of reference (see Table EV6 for details regarding used datasets). Parameters were set to "mean.only = T" and "par.prior = T". Each study was defined as a different batch where the control condition (i.e., non-injured livers) was matched to the adult liver stage of the reference dataset. Next, a PCA was computed only on the mouse liver differentiation study with the "PCA" function of FactomineR (Lê *et al*, 2008; using "scale.unit = F"). Liver injury studies were considered as supplemental individuals. Finally, the first principal component (representing 63.55% of the variability of the mouse liver differentiation study) was plotted and used to project the liver injury studies. Data corresponding to prenatal mouse livers were used in the analyses but were discarded for data visualization.

### Functional enrichment analyses

Functional enrichment analyses were performed using the Topp-Gene Suite (Chen *et al*, 2009). KEGG Pathways with Bonferroni-corrected $P < 10^{-3}$ and Gene Ontology (GO) Biological Processes with Bonferroni-corrected $P < 10^{-6}$ were considered, and similar terms were merged.

### Gene set enrichment analyses

Gene set enrichment analyses (GSEA) were performed using the GSEA software (v3.0) developed at the Broad Institute (Subramanian *et al*, 2005). We used 1,000 gene-set permutations and the following settings: "weighted" as the enrichment statistic and "difference of classes" as the metric for ranking genes. Ranking was performed by the GSEA software using the average expression value per gene when multiple probesets were present in the microarray. In addition to enrichment plots, figures also provide NES and FDR, which are the normalized enrichment score and the false discovery rate provided by the GSEA software, respectively. In experiments with multiple conditions, the BubbleGUM tool (GSEA Unlimited Map v1.3.19; Spinelli *et al*, 2015) was used to integrate and compare numerous GSEA results with multiple testing correction. Non-oriented GO term enrichment analyses were performed using the "MousePath_GO_gmt.gmt" set of genes from the Gene Set Knowledgebase (GSKB; preprint: Lai *et al*, 2016).

### Chromatin immunoprecipitation

MPH ($3 \times 10^6$ cells) were fixed for 30 min at room temperature with disuccinimidyl glutarate followed by a 10-min incubation with 1% formaldehyde and a 5-min incubation with 125 mM glycine. After two washes with ice-cold PBS, cells were scraped in PBS, pelleted by centrifugation at 400 *g* for 5 min, resuspended in Lysis Buffer (50 mM Tris–HCl pH 8.0, 10 mM EDTA, 1% SDS, and 1× PIC from Roche), and sonicated for 4 min (four cycles 30 s ON/30 s OFF using Bioruptor Pico from Diagenode). Mouse liver (200 mg of tissue) was cut in small pieces in ice-cold PBS, pressed through a 70-μm cell strainer followed by a few passages through a 18G needle. Fixation, lysis, and sonication were performed as described for MPH. Chromatin (50 μg for H3K27ac ChIP and 200 μg for BRD4

ChIP) was diluted 10-fold in Dilution Buffer (20 mM Tris–HCl pH 8.0, 1% Triton X-100, 2 mM EDTA, 150 mM NaCl) and incubated overnight with 2 µg of H3K27ac antibody (Active Motif, #39685) or 3 µg of BRD4 antibody (Bethyl Labs, #A301-985A100) at 4°C. The next day, A/G sepharose bead mix (GE Healthcare) was added during 4 h at 4°C in the presence of 70 µg/ml yeast tRNA (Sigma-Aldrich). Beads were washed three times with RIPA buffer (50 mM HEPES pH 7.5, 1 mM EDTA, 0.7% Na deoxycholate, 1% NP-40, and 500 mM LiCl) containing 10 µg/ml yeast tRNA and once with TE buffer (10 mM Tris–HCl pH 8.0, 1 mM EDTA). DNA was then eluted in 100 mM NaHCO$_3$ containing 1% SDS and incubated overnight at 65°C for reverse-crosslinking. DNA purification was performed using the MinElute PCR purification kit (Qiagen, #2800), and samples were subjected to qPCR analyses. The primer sequences are listed in Table EV5.

H3K27ac ChIP and input samples from MPH treated for 4 h with 1 µM thapsigargin or vehicle (0.04% DMSO) from three independent experiments were additionally sent for sequencing on an Illumina Hi-seq 4000 as single-end 50-base reads according to the manufacturer's instructions. Data were analyzed as described hereafter and have been submitted to GEO under accession number GSE122508.

### ChIP-seq data analyses

ChIP-seq data quality control and uniform reprocessing including mapping to the mm10 version of the mouse genome and signal normalization have been described in Dubois-Chevalier et al (2017) except Bowtie 2 (sensitive mode; Langmead et al, 2009) was used for the BRD4 ChIP-seq analyses. ChIP-seq data were visualized using the Integrated Genome Browser (IGB 9.0.1; Freese et al, 2016).

### Broad H3K4me3 domain identification and identity gene definition

H3K4me3 ENCODE ChIP-seq data from several mouse tissues (Shen et al, 2012; Table EV6) were used to call broad H3K4me3-enriched regions using MACS2 as described in Chen et al (2015). Broad H3K4me3 domains were defined as those spanning more than three times the median size of all H3K4me3-enriched regions in a given tissue. Broad H3K4me3 domains from mouse liver were separated into liver-identity (LIVER-ID) domains, which were defined as broad H3K4me3 domains specific to liver (i.e., detected in < 25% of other analysed tissues), and in ubiquitous (UBQ) domains. LIVER-ID and UBQ domains were then assigned to genes according to overlapping TSS from the GENCODE (M9) database (Frankish et al, 2019), resulting in 621 LIVER-ID genes and 657 UBQ genes which are listed in Table EV1. TFs within these gene lists were subsequently obtained using comparison with mouse TFs listed in the Animal TFDB 2.0 (Zhang et al, 2015). Muscle-identity (MUSCLE-ID) and UBQ genes, listed in Table EV4, were defined in a similar way using H3K4me3 ChIP-seq data from the ROADMAP consortium processed by Chen et al (2015). Human to mouse gene name conversion was performed using the dbOrtho tool from bioDBnet (Mudunuri et al, 2009).

### Super-enhancer identification

To define BRD4 super-enhancers (SE), we first used MACS2 to identify enriched peaks (effective genome size = 2150570000, bandwidth = 300, mfold = 5–50, FDR (q-value) = 0.05, max duplicate tags at the same location = 1) using mapped reads previously filtered to remove duplicates and reads mapping to false positives regions we had identified in Dubois-Chevalier et al (2017). SE were identified by applying rank ordering of super-enhancers (ROSE; Loven et al, 2013; Whyte et al, 2013) on the BRD4 peak-calling results using mouse liver ChIP-seq inputs (GSE26345) as control (setting: -s 12500, -t 0).

### Identification of changes in H3K27ac induced by acute ERS

Regions with significant changes in H3K27ac ChIP-seq signals induced by ERS were identified using csaw 1.6.1 (Lun & Smyth, 2014, 2016). Mapped reads were previously filtered to remove duplicates and reads mapping to ENCODE blacklisted regions (Encode_Project_Consortium, 2012) or mouse ChIP-seq false-positive regions we had identified in Dubois-Chevalier et al (2017). The command lines and full list of used parameters are provided in Computer Code EV1. Briefly, the genome was binned and reads counted, bins with background level signal as defined using input samples were discarded before normalization using a loess regression. Finally, after dispersion estimation with the function estimateDisp, a paired-differential analysis was performed on this filtered and normalized data using glmQLFit. Bins overlapping H3K27ac peaks (broad regions called with MACS2 using a pool of all H3K27ac ChIP-seq datasets and inputs as control—parameters: q-val narrow = 0.001 and q-val broad = 0.01) were identified using findOverlaps from GenomicRanges 1.24.3 (Lawrence et al, 2013; parameters : minoverlap = 75, maxgap = 0). Bins overlapping a single H3K27ac peak were combined using combineOverlaps, and only merged bins with FDR ≤ 0.05 were considered (merged bins with FDR > 0.05 were defined as unchanged H3K27ac regions). The ratio of UP to DOWN bins in the merged regions was next calculated, and H3K27ac UP or DOWN regions were defined as those having a ratio ≥ 2 or ≤ 0.5, respectively. Coordinates for H3K27ac UP, DOWN, and unchanged regions are provided in Dataset EV1. The bigwig signals were computed using the loess normalized signal on each dataset and/or averaging the loess normalized signal between replicates. Genes were assigned to H3K27ac regions as follows: First, genes whose TSS from the GENCODE (M9) database (Frankish et al, 2019) directly overlaps H3K27ac regions were retrieved. In addition, distal H3K27ac was linked to potentially regulated genes using CisMapper (O'Connor et al, 2017) as previously described in (Dubois-Chevalier et al, 2017).

### Analyses of transcriptional regulators recruited to regions with changes in H3K27ac induced by acute ERS

In order to identify TFs whose binding is enriched in H3K27ac UP and H3K27ac DOWN regions, we used Locus Overlap Analysis (LOLA 1.4.0; Sheffield & Bock, 2016) to compare TF binding within UP, DOWN, and ALL (i.e., also including H3K27ac unchanged) regions. Mouse TF-binding sites were retrieved from the Gene Transcription Regulation Database (GTRD) (Metaclusters of GTRD release 16.07; Yevshin et al, 2017). Inputs were discarded, and ChIP-seq datasets were ascribed to TFs using nomenclature information provided by the authors. A heatmap of log-odds ratio was generated using the heatmap.2 function of the R package "gplots" (v3.0.1; Warnes et al, 2016) and hierarchical clustering using the hclust function of the R package "Stats" (using Euclidean distance

and ward.D2 agglomeration method; R Core Team, 2015). A list of the TFs of each cluster is shown in Table EV2.

H3K27ac UP, DOWN, or unchanged regions were overlapped with *cis*-regulatory modules (CRMs) defined in Dubois-Chevalier *et al* (2017) based on co-binding of 47 transcriptional regulators in mouse liver. Combinatorial co-binding of transcriptional regulators at H3K27ac UP, DOWN, or unchanged was analyzed using multidimensional scaling (MDS) analyses as described in Dubois-Chevalier *et al* (2017). Plots were performed with the smoothScatter function of the «graphics» R package (R Core Team, 2015) using a conserved color scale.

### Public transcriptomic and functional genomic data recovery

Public data used in this study were downloaded from Gene Expression Omnibus (GEO; http://www.ncbi.nlm.nih.gov/geo/; Edgar *et al*, 2002), ENCODE (Yue *et al*, 2014), UCSC Genome Browser (Dreszer *et al*, 2012), or BioGPS (Mouse MOE430 Gene Atlas; Wu *et al*, 2016) and are listed in Table EV6.

### Human samples

Postmortem liver biopsies from patients admitted to the intensive care unit (ICU) of Leuven University Hospital with sepsis ($n = 64$), who died after a median ICU stay of 10 days (IQR 6–20 days), were compared with matched patients undergoing elective restorative rectal surgery ($n = 18$). Written informed consent was obtained from the patients or their closest family member and from the volunteers. The study protocols and consent forms were approved by the KU Leuven Institutional Review Board (ML1094, ML1820, and ML2707). Bilirubin was quantified with the use of a standard routine automated assay in the University Hospital Clinical Laboratory.

### Protein extraction
#### Total extracts

MPH and AML12 cells were scraped in ice-cold PBS, pelleted by centrifugation at 400 $g$ for 5 min, lysed in Laemmli buffer 6× (175 mM Tris–HCl pH 6.8, 15% glycerol, 5% SDS, 300 mM DTT, and 0.01% Bromophenol Blue) and sonicated for 10 min. Mouse liver was cut in small pieces in ice-cold PBS and pressed through a 70-μm cell strainer. The pellet obtained after centrifugation at 400 $g$ for 5 min was lysed and sonicated as described for MPH. Western blottings shown in this study were obtained using total cellular extracts unless indicated otherwise.

#### Nuclear extracts

MPH were scraped in ice-cold PBS, and mouse liver was cut in small pieces in ice-cold PBS and pressed through a 70-μm cell strainer. Pellets were obtained by centrifugation at 400 $g$ for 5 min, lysed in Hypotonic Buffer (20 mM Tris–HCl pH 8.0, 10 mM NaCl, 3 mM $MgCl_2$, 0.2% NP-40, and 1× PIC from Roche) and incubated for 5 min at 4°C. Samples were centrifuged at 600 $g$ for 5 min at 4°C and supernatants constituted the cytoplasmic fraction. Nuclear pellets were lysed in Nucleus Lysis Buffer (25 mM Tris–HCl pH 8.0, 500 mM NaCl, 1 mM EDTA, 0.5% NP-40, and 1× PIC from Roche). Hypotonic Buffer and Nucleus Lysis Buffer were supplemented with Phosphatase Inhibitor Cocktail (#P044 from Sigma-Aldrich) as well as with 5 mM Sodium Butyrate and 5 μM Trichostatin A for deacetylase inhibition. After incubation for 30 min at 4°C, samples

were sonicated for 10 min and centrifuged at 16,000 $g$ for 5 min at 4°C. Laemmli 6× was added to the supernatants which were used for Western immunoblotting.

#### Chromatin fraction

MPH were scraped in ice-cold PBS, pelleted by centrifugation at 400 $g$ for 5 min, lysed in Buffer A (50 mM HEPES pH 7.5, 10 mM KCl, 1.5 mM $MgCl_2$, 340 mM sucrose, 10% glycerol, 1 mM DTT, and 1× PIC from Roche), and incubated for 10 min at 4°C. Samples were centrifuged at 1,300 $g$ for 5 min at 4°C, and supernatants were discarded. Nuclear pellets were washed with Buffer A and subsequently lysed in solution B (3 mM EDTA, 0.2 mM EGTA, 1 mM DTT, and 1× PIC from Roche). After incubation for 30 min at 4°C, samples were centrifuged at 1,700 $g$ for 5 min at 4°C and supernatants were discarded. Chromatin pellets were washed with solution B, resuspended in Buffer C (50 mM Tris–HCl pH 8.0, 1 mM $MgCl_2$, and 83 U/μl benzonase), and incubated for 20 min at 4°C. Laemmli buffer 6× was added before loading for Western immunoblotting.

### Plasmids and *in vitro* transcription and translation

The pcDNA3.1-mNFIL3 (Addgene 34572) and pSGG5-hHNF4A constructs were used for *in vitro* transcription and translation (*in vitro* TNT) using the TnT® Quick Coupled Transcription/Translation System (Promega).

### Western immunoblotting
#### Western blot assays (WB)

One hundred μg of proteins was separated by 10% SDS-PAGE and immunodetected by Western immunoblotting using the primary antibodies listed in the Reagents and Tools table. Primary antibodies were detected using HRP-conjugated secondary antibodies (Sigma-Aldrich). Images were acquired using a G-box (Syngene, Cambridge, UK) or using the iBright™ CL1500 Imaging System (Thermo Fisher Scientific). Quantifications were performed using Image Studio Lite v5.2 (LI-COR Biosciences, Lincoln, USA), and band intensities were defined using the signal value (sum of the pixel intensity in a shape minus the background value).

#### Simple Western immunoassays (WES)

Simple Western size-based assays were run on a WES system as recommended by the manufacturer (ProteinSimple, San Jose, USA). Protein concentrations ranged from 0.25 to 0.8 μg/μl depending on the target protein. Primary antibodies are listed in the Reagents and Tools table. Secondary antibodies were provided by the manufacturer (PS-MK14 and PS-MK15, ProteinSimple). Data were analyzed using the Compass software (ProteinSimple). Quantifications were obtained using the area under the peak of the protein of interest.

### Co-immunoprecipitation assays

MPH cells were resuspended into Hypotonic Buffer (20 mM Tris–HCl, pH 7.5, 10 mM NaCl, 3 mM $MgCl_2$, 0.2% NP-40, and protease inhibitors), and the pellet was lysed for 30 min. After 10-min sonication (30-s on/off cycles with a Bioruptor (Diagenode) and centrifugation, 500 μg nuclear proteins from the soluble fraction were diluted with two volumes of a buffer containing 25 mM Tris–HCl pH7.5, 1 mM EDTA, 1.5 mM $MgCl_2$, and incubated overnight

with 2 μg of p300 antibody (Active motif, #61401) or control mouse IgG (sc-2025, Santa Cruz). Samples were then incubated for 4 h with magnetic beads (Life technologies) previously blocked with 5 mg/ml of serum albumin bovine and washed 4 times using ice-cold washing buffer containing 25 mM Tris–HCl pH7.5, 150 mM NaCl, 1 mM EDTA, 0.2% NP-40, and protease inhibitors. Beads were finally eluted in Laemmli buffer 6×.

### Statistical analyses

Statistical analyses were performed using the Prism software (GraphPad, San Diego, CA) and R (R Core Team, 2015). The specific tests and corrections for multiple testing which were used as well as the number of samples per condition are indicated in the figure legends. In all instances, statistical significance was considered to be reached when $P$-values were below 0.05, which was indicated by * or #. All bar graphs show means $\pm$ SD (standard deviations). Box plots are composed of a box from the 25th to the 75th percentile with the median as a line and min to max as whiskers.

## Data availability

The datasets produced in this study are available in the following databases:

- Chip-Seq data: Gene Expression Omnibus GSE122613 (https://www.ncbi.nlm.nih.gov/geo/query/acc.cgi?acc=GSE122613)
- Transcriptomic data: Gene Expression Omnibus GSE122508 (https://www.ncbi.nlm.nih.gov/geo/query/acc.cgi?acc=GSE122508).

**Expanded View** for this article is available online.

## Acknowledgements

The authors acknowledge O. Molendi-Coste, A. Grenon, F. Firmin, C. Mazuy, X. Maréchal, A. Berthier, D. Sanchez-Lopez, O. Briand, B. Derudas, J. Vandel, L. L'homme (INSERM U1011, Lille, France), Lieselotte Thoolen (Amsterdam Animal Research Center, VU University Amsterdam, The Netherlands), and Shwu-Yuan Wu (University of Texas Southwestern Medical Center, Dallas, USA) for technical assistance and J. Dubois for administrative support. The authors also thank Lei Yin (University of Michigan Medical School, USA), L. Dubuquoy (INSERM U995, Lille, France), and Lies Langouche (KU Leuven, Belgium) for helpful discussions. This work was supported by grants from the Fondation pour la Recherche Médicale (Equipe labellisée, DEQ20150331724), "European Genomic Institute for Diabetes" (E.G.I.D., ANR-10-LABX-46) and European Commission. B.S. is supported by the European Research Council (ERC Grant Immunobile, contract 694717). C-M.C. is supported by US National Institutes of Health (NIH 1RO1CA251698-01), Cancer Prevention & Research Institute of Texas (CPRIT RP180349 and RP190077), and Welch Foundation (I-1805).

## Author contributions

Conceptualization: VD, PL, BS, and JE; Methodology: VD, PL, BS, JE, RP, GVB, IV, HDu, JS-A, EW, EB, FL, and MB-G; Software: JD-C and FG; Validation: VD, WV, CG, JD-C, HDe, MV, and MB-G; Formal Analysis: VD, JD-C, JE, and FG; Investigation: VD, WV, CG, JD-C, HDe, MV, FPZ, MP, LP, and MB-G; Resources: NR, JS-A, AB, CD, SL, HDu, REK, C-MC, IV, GV, RP, ED, and EV; Data Curation: VD, JD-C, and JE; Writing—Review & Editing: VD and JE with inputs from co-authors; Visualization: VD and JD-C; Project Administration: JE; Funding Acquisition: JE, PL, and BS.

## Conflict of interest

BS is consultant for Genfit SA. The remaining authors declare no competing financial interest.

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
