## [Review Process File · Molecular Systems Biology]

Endoplasmic reticulum stress actively suppresses hepatic molecular identity in damaged liver

Vanessa Dubois, Céline Gheeraert, Wouter Vankrunkelsven, Julie Dubois-Chevalier, Helene Dehondt, Marie Bobowski-Gerard, Manjula Vinod, Francesco Zummo, Fabian Güiza, Maheul Ploton, Emilie Dorchies, Laurent Pineau, Alexis Boulinguez, Emmanuelle Vallez, Eloise Woittrain, Eric Baugé, Fanny Lalloyer, Christian Duhem, Nabil Rabhi, Ronald van Kesteren, Cheng-Ming Chiang, Steve Lancel, Hélène Duez, Jean-Sébastien Annicotte, Réjane Paumelle, Ilse Vanhorebeek, Greet Van den Berghe, Bart Staels, Philippe Lefebvre and Jerome Eeckhoutte.

Review timeline:

Submission date:	24 th October 2019
Editorial Decision:	26 th November 2019
Revision received:	25 th February 2020
Editorial Decision:	7 th April 2020
Revision received:	9 th April 2020
Accepted:	14 th April 2020

Editor: Maria Polychronidou

Transaction Report:

1st Editorial Decision

26th November 2019

Thank you again for submitting your work to Molecular Systems Biology. We have now heard back from the three referees who agreed to evaluate your study. As you will see below, the reviewers acknowledge that the presented data and mechanistic insights seem potentially interesting. They raise however a series of concerns, which we would ask you to address in a major revision.

Without repeating all the points listed below, some of the more fundamental issues are the following. Reviewers #1 and #2 point out that several of the presented conclusions remain somewhat correlative and think that further validations would be required to provide more convincing support. Both reviewers include constructive suggestions on further analyses that could be performed. We do not think that it is necessary to remove data from the manuscript, but we would ask you to make sure that they are described in a coherent manner.

All other issues raised by the reviewers would need to be addressed. Please feel free to contact me in case you would like to discuss in further detail any of the issues raised.

REFeree REPORTS

Reviewer #1:

REVIEWER COMMENTS

In this manuscript, the authors proposed that the response to endoplasmic reticulum (ER) stress affects the transcriptome landscape in damaged liver by suppressing hepatic molecular identity. The authors demonstrated that the unfolded protein response (UPR) transcriptional program is upregulated while the master liver transcriptional regulators and effector genes (referred as LIVER-ID) are downregulated upon treatment with ER stress inducing drugs tunicamycin in mouse primary hepatocytes (MPH) and mice models. Next, they identified BRD4 super-enhancers (SE) to be repressed the expression of LIVER-ID genes upon ER stress. The decrease in LIVER-ID genes is proposed to be mediated by the proteasome-dependent degradation of LIVER-ID transcription factors, promoted by the proto-oncogene tyrosine-protein kinase Src. Furthermore, they identified ER stress-mediated upregulation of NFIL3 gene to further attenuate LIVER-ID genes in mice. Finally, they proposed that the ER stress-mediated LIVER-ID gene suppression occurs in dysfunctional mouse and human livers.

Overall, the manuscript contains a large amount of data. Although I am not an expert of liver dysfunctions and transcription regulation through cis-regulatory modules (CRMs) and super-enhancer, but I feel that of the finding presented are not well connected to each other and are possibly only correlative. For instance, the findings related to NFIL3 are interesting, but I have a hard time to consolidate these observations with the rest of the story. It becomes obviously problematic looking at the proposed model in Fig. 6 where the authors connect their different observations with "arrows" but how "a" connects to "b" and "b" to "c" is a big question mark. As the manuscript contains a lot of data, I found it difficult to grasp at time due to the lack of details so I had to go back to the methods, I had to search the functions of some of the genes/proteins part of the study, to the generalized use of the jargon in the field. Perhaps, the authors should consider removing some of the data that are not strongly supporting the conclusion to eventually come up with a simplified but strong revised model. I believe the manuscript will then be more appealing and compressive. It will also leave rooms to add some important controls and validation that are missing in my opinions. Again, based on my expertise, most of the comments below will focus on ER stress and the UPR.

Major Points

1. My major issue with the manuscript is that there is no validation that their observations are directly related to ER stress response (unfolded protein response, UPR). Someone needs to block the UPR (IRE1 α , PERK, ATF6) genetically or by using commercially available inhibitors to directly demonstrated the role of the UPR in remodelling the expression of LIVER-ID genes. This is rather important as the authors used the drug tunicamycin which blocks the glycosylation of all ER proteins which in turn might affect other processes like protein trafficking, endocytosis, autophagy. Finally, during the acute phase of ER stress, there is a general but temporary inhibition of translation mediated by PERK through the phosphorylation of eIF2 α . This is never explored nor mentioned. I suspect that it might play a role in the observed remodelling of transcription factors as well as the change in LIVER-ID genetic expression.
2. As mentioned above, I don't see the connection between some of the results and are at best correlative with the current provided evidences. In my view, the manuscript will be more impactful and attractive if the authors decide to remove some of the distracting data (can the manuscript be separated into 2?). It will also leave room to add key data that will directly link the different observations with strong evidence including what is described in "Major point #1".
3. I find it very unconventional to use the term ERS (ER stress) repeatedly to describe the upregulated transcriptional program that is modulated by the unfolded protein response (UPR). The UPR is activated by the ER stress transducers IRE1 α , PERK, ATF6. The authors should refer to the UPR through the manuscript. I think a short introduction on the UPR pathway is needed as it is central to the story. The UPR (referred to ERS by the authors) seems to be a black box even in the proposed model in Fig. 6.
4. The detection levels of p-Y416-SRC and total SRC are quite low. The loading control is also of poor quality. There are many examples of strong p-Y416-SRC detection in the literature (PMID 15546918, 22411953, 23145001). It needs to be revised.

5. I find it strange to have some of the results description buried into the supplementary file (additional results). I think it should be moved to the main text unless the journal policies allow it.

Minor points

There are some minor points that should be considered below before publication if they haven't been already addressed by the authors.

Minor Points

1. Page 5, replace "These hepatic ERS UP gens were" by "These hepatic ERS UP genes were".
2. Page 16, there is a parenthesis missing after "expression (events occurring in MPH in the absence...".
3. I dislike the use of the term "4h ERS" instead of "4h Tm". I think it is really misleading especially when put together in a figure together with inhibiting drug like MG132. Please not that by convention we use the term Tm instead of Tn for tunicamycin. This should be corrected through the manuscript and the figures.

Reviewer #2:

In the manuscript "Endoplasmic reticulum stress actively suppresses hepatic molecular identity in damaged liver", Vanessa Dubois and colleagues address a well-established phenomenon: Hepatic endoplasmic reticulum stress (ERS) represses the expression of a set of genes that determine liver functions. The analysis of published transcriptomic and genomic data is nicely combined with new, well-designed experiments whereby the authors characterize the inhibition of liver identity genes under diverse ER stress conditions (ERS). To that aim the authors take advantage of several experimental systems where ERS is induced.

From the transcriptomic analysis of thapsigargin-treated primary hepatocytes, Dubois and colleagues define sets of ERS-induced and ERS-repressed genes. In agreement with previous observations, a significant fraction of ERS downregulated genes encodes factors that support the metabolic functions of differentiated hepatocytes; these genes are identified in this manuscript as LIVER ID genes. Under non-ERS conditions, the expression of LIVER ID genes is robust and (to a large extent) hepatospecific. Epigenetically, this set of genes is featured by broad H3K4me3 histone marks at their promoters, and high levels of H3K27 acetylation, as well as by the association of a node of multiple hepatospecific transcription factors (LIVER ID TFs) that, in turn, recruit BRD4 into superenhancer domains. This work documents the molecular alterations whereby LIVER ID genes are repressed under ERS: 1) the enhancer H3K27ac mark decreases (while it increases in ERS up genes), 2) BRD4 is decommissioned from these promoters, and 3) the level of several LIVER ID TFs is reduced. Nicely enough, the analysis of ChIPseq data confirmed that robustness of LIVER ID gene expression is established by the coordinated binding of multiple liver IDs TFs to LIVER ID genes. This manuscript identifies two mechanisms, 1) the degradation of LIVER ID TFs through an SRC-dependent mechanism or 2) the inhibition of specific LIVER ID TFs by the ERS-induced transcriptional inhibitor NFIL3 in the inhibition of part of the hepatic gene expression program.

Together, the evidences presented in this manuscript integrate previous and new observations into a model that, again, is coherent with earlier published data; the novelty of some findings is relative. On the other hand, this work proposes specific molecular mechanisms that would account for LIVER ID gene repression. While these mechanisms may explain how liver transcription is tuned under ER stress, the experimental evidence supporting them is sometimes insufficient or not precise enough and it should be revised before publication.

Major points:

- The authors document that the reduction of LIVER ID mRNAs levels under ERS is stronger than the decrease of "ubiquitous" genes (featured by the presence of broad H3K4me3 marks at their promoters) or "other genes"; thus, it is tempting to assume that the mechanism of LIVER ID gene

inhibition is selective. Alternatively, the extent of transcriptional inhibition could be proportional to initial/steady-state expression levels in such a way that it could be easier to score in highly expressed (i.e. LIVER ID) genes. To rule out that possibility, the transcriptional downregulation of LIVER ID and ubiquitous genes having similar basal expression levels should be compared.

- The experiments performed to address the decommissioning of BRD4 from the LIVER ID genes convincingly show that, under ER stress, the decreased association of BRD4 with LIVER ID promoters correlates with lower H3K27 acetylation of these genes. Based on these observations, as well as on the ChIP-PCR analysis of BRD4 association to LIVER ID genes (Figure 2G), this manuscript proposes that BRD4 is redistributed from LIVER ID to ERS genes where it could convey higher transcription rates. Unlike LIVER ID genes, ERS genes are not really sensitive to BRD4 inhibition and therefore the redistribution of BRD4 may not be relevant for ERS transcriptional activation. While many of the evidences supporting the correlation between LIVER ID binding, BRD4 recruitment and H3K27 acetylation are in good agreement with the main conclusions of this work, there are many instances where more/better evidence is needed.

- a. In the Western blot of Fig. 2F BRD4 protein levels could be reduced by ER stress; the long band seem to be saturated in the control samples, but ERS may reduce the signal of the long band. This ERS-induced reduction seems a bit more clear in the MZ1+ERS samples (when compared to cells treated with MZ1 alone).

- b. The ChIP-PCR analysis shown in Fig. 2G represents genomic domains corresponding to 8 ERS genes and 5 LIVER-ID TF genes, which were analyzed in Figs. S7 and S8. Given the low number of genes/genomic locations analyzed, it would be good to show the data corresponding to each individual ChIP-PCR (as Supplementary data) rather than an average of all measurements.

- c. The correlation between LIVER ID TF binding to LIVER ID genes, the extent of H3K27 acetylation and the recruitment of BRD4 depicts the molecular features of LIVER ID genes (and the loss of these features characterizes their inhibition); but, how are these molecular features functionally related with each other in the context of ERS-dependent transcriptional inhibition? Can we establish their contribution to LIVER ID downregulation more precisely? For instance, Trichostatin A suppresses the inhibition of LIVER ID genes under ERS, but does it prevent the rapid degradation of LIVER ID TFs? Is BRD4 inhibition sufficient to promote LIVER ID TF degradation? How plausible is it that the destabilization of LIVER ID promoters/enhancers results from the inhibition of one of the events considered in this study?

- The levels of some of LIVER ID transcription factors are strongly reduced by ER stress. The experimental evidence nicely documents that in thapsigargin-treated hepatocytes HNF4A, NR1H4 and FOXA2 protein levels drop even before mRNA levels change in response to ERS. The authors state that this reduction is due to the enhanced proteasomal degradation of these TFs, since addition of MG132 yields higher TF levels. However, when compared with the cells treated only with MG132, the combined thapsigargin+MG132 also reduces the levels of FOXA2 or HNF4A (and even of NR1H4), indicating that this phenomenon still occurs when proteasomes are inhibited. In the absence of a quantitation of these Western blots (that the authors should include), it seems clear that proteasomal degradation of these factors does not account for the downregulation of LIVER ID TFs observed. Similarly, the treatment of stressed cells with PP2 does not fully prevent the dampening of LIVER ID TFs.

- Since proteasomal degradation may not be the main (or the only) driver of LIVER ID TF decay, alternative mechanisms should be considered. For instance, the translational inhibition imposed by the ERS transducer PERK could prevent HNF4A/FOXA2 translation and thereby contribute to their downregulation. Pharmacological inhibition of PERK signaling using ISRIB or GSK2606414 could address this possibility.

- Along with this idea, it is somewhat surprising that the potential link between the three independent UPR mechanisms and the inhibition of LIVER ID genes has not been explored to some detail in this work. In fact, SRC-dependent degradation of TFs depends on IRE1. NFIL3 has been identified as a potential ATF6 target gene, while CHOP expression is controlled by the three UPR signaling mechanisms but maybe most importantly by PERK. Therefore, it would be in the scope of this work to determine how/which UPR branch (if any) mediates the downregulation of LIVER ID genes.

- Is sepsis a good surrogate of ER stress? How specific is the link between ERS and LIVER ID downregulation? Elegantly, the authors demonstrate ERS is resolved after hepatectomy in wild-type but not in ATF6-deficient mice. Unresolved ER stress leads to a prolonged inhibition of LIVER ID genes. This observation establishes a nice functional correlation between UPR/ERS and liver ID gene expression.

But, would any type of liver damage cause the same downregulation of liver identity genes? Is

sepsis causing this "loss of identity" through ERS or could it be linked to a more unspecific, general stress response? PBA or TUDCA have been shown to alleviate ER stress signaling in the liver of mice; while the mechanism by which these drugs relieve ER stress is still unclear, the authors may use these compounds to evaluate their capacity to prevent (or diminish) LIVER ID downregulation in sepsis models. Other than that, it would be interesting to test if hepatic insults not linked to ERS could promote the decreased expression of liver identity genes.

Minor points:

- In Fig. 1C, the authors claim that PHx and ERS are associated with more complex and widespread transcriptomic alterations than, for instance, the physiological fast/fed metabolic shift. This comparative analysis was performed to determine if downregulation of the hepatic gene expression program under acute stress might occur as a consequence of the exceptional requirement for induction of novel genes. Although the analysis documents nicely the amplitude of transcriptomic responses under ERS, it does not address properly/solve this relevant question.
- The effect of NFIL3 downregulation in xenobiotic metabolism genes is modest, but still significant. The authors should provide a table including the list of genes that are less downregulated by ERS in the NFIL3 ko as well as the extent of downregulation and the functional category to which they belong.
- The variability of data in Fig. 5 M is really high. It is hard to draw any conclusion from these data. Of note, CHOP/DDIT3 expression could result from the activation of eIF2alpha kinases other than PERK (i.e. PERK, HRI) that would be very likely activated in septic livers, so CHOP expression is not the best surrogate of ERS/ERS responses.

Reviewer #3:

In this paper Dubois et al describe multiple mechanisms responsible for endoplasmic reticulum stress-mediated gene expression changes in the liver. The authors compared hepatic gene expression patterns in livers following partial hepatectomy, in primary hepatocytes treated with endoplasmic reticulum stress-inducing agents as well as in KO mice lacking NFIL3 or HNF4 and in livers from mouse sepsis models.

The main conclusion of the study is that liver injury-mediated downregulation of hepatocyte identity genes involves reduction of key hepatic transcription factor protein levels, via increased degradation and via decreased expression. It is demonstrated that the effect on protein degradation involves Src3 kinase activation, while the transcriptional effect is mediated by decommissioning Brd4 at superenhancers that are densely co-bound by hepatic TFs.

The above findings are novel and represent a significant advance to our current knowledge. They are based on a comprehensive analysis of existing and newly generated ChIP-seq end expression profiling data sets. The analysis seems very thorough, in most cases involving parallel, independent approaches, which strengthen the conclusions. The main mechanistic findings are also supported by the analyses of data from mouse models where sepsis is induced and from livers from deceased septic patients.

The study represents an important reference work, which is expected to be heavily used by other investigators.

Points to be considered before acceptance:

1) The regulatory role of NFIL3 induction in the mechanism of ERS-mediated repression of hepatic genes, seems to be overemphasized. While the presented data showing that NFIL3 expression is increased and that the expression of some of the ERS-repressed genes are less affected in NFIL3 KO mice are clear, the overall extent of changes seem quite marginal, arguing against of a major role of NFIL3 induction in the observed effects. This part of the results does not substantially affect the story and could either be removed or NFIL3 should be discussed as a minor potential contributor in the mechanism.

2) The analysis in Figure 5D showing correlation of gene expression changes in ERS induced primary hepatocytes and septic livers is clear. It would be informative to show the number of

overlapping genes in venn diagrams.

3) It is known and also shown in Figure 1 that following ERS induction or other liver injuries, hepatic gene expression patterns after the initial drop recover to normal. The mechanism of recovery should be more explicitly discussed in the text by explaining the scheme of Figure 6.

We would like to thank the reviewers for their insightful comments. We provide below a point-by-point response to the reviewers' criticisms and suggestions for improvement. We also provide a marked version of the revised manuscript where main changes are highlighted in red and line numbers have been added.

Reviewer #1

In this manuscript, the authors proposed that the response to endoplasmic reticulum (ER) stress affects the transcriptome landscape in damaged liver by suppressing hepatic molecular identity. The authors demonstrated that the unfolded protein response (UPR) transcriptional program is upregulated while the master liver transcriptional regulators and effector genes (referred as LIVER-ID) are downregulated upon treatment with ER stress inducing drugs tunicamycin in mouse primary hepatocytes (MPH) and mice models. Next, they identified BRD4 super-enhancers (SE) to be repressed the expression of LIVER-ID genes upon ER stress. The decrease in LIVER-ID genes is proposed to be mediated by the proteasome-dependent degradation of LIVER-ID transcription factors, promoted by the proto-oncogene tyrosine-protein kinase Src. Furthermore, they identified ER stress-mediated upregulation of NFIL3 gene to further attenuate LIVER-ID genes in mice. Finally, they proposed that the ER stress-mediated LIVER-ID gene suppression occurs in dysfunctional mouse and human livers.

Overall, the manuscript contains a large amount of data. Although I am not an expert of liver dysfunctions and transcription regulation through cis-regulatory modules (CRMs) and super-enhancer, but I feel that of the finding presented are not well connected to each other and are possibly only correlative. For instance, the findings related to NFIL3 are interesting, but I have a hard time to consolidate these observations with the rest of the story. It becomes obviously problematic looking at the proposed model in Fig. 6 where the authors connect their different observations with "arrows" but how "a" connects to "b" and "b" to "c" is a big question mark. As the manuscript contains a lot of data, I found it difficult to grasp at time due to the lack of details so I had to go back to the methods, I had to search the functions of some of the genes/proteins part of the study, to the generalized use of the jargon in the field. Perhaps, the authors should consider removing some of the data that are not strongly supporting the conclusion to eventually come up with a simplified but strong revised model. I believe the manuscript will then be more appealing and compressive. It will also leave rooms to add some important controls and validation that are missing in my opinions. Again, based on my expertise, most of the comments below will focus on ER stress and the UPR.

Major Points

1. My major issue with the manuscript is that there is no validation that their observations are directly related to ER stress response (unfolded protein response, UPR). Someone needs to block the UPR (IRE1 α , PERK, ATF6) genetically or by using commercially available inhibitors to directly demonstrated the role of the UPR in remodelling the expression of LIVER-ID genes. This is rather important as the authors used the drug tunicamycin which blocks the glycosylation of all ER proteins which in turn might affect other processes like protein trafficking, endocytosis, autophagy. Finally, during the acute phase of ER stress, there is a general but temporary inhibition of translation mediated by PERK through the phosphorylation of eIF2 α . This is never explored nor mentioned. I suspect that it might play a role in the observed remodelling of transcription factors as well as the change in LIVER-ID genetic expression.

Throughout our study, we provide several experimental evidences that repression of LIVER-ID genes is due to ER stress, which are now listed and discussed p.15 lines 24-26 and p.16 lines 1-12. First, ruling out any drug-specific artifact (such as those listed by the reviewer) as being the trigger for the LIVER-ID gene downregulation, we showed that this repression occurs when hepatocytes are treated both with tunicamycin and thapsigargin (e.g. Appendix Fig.S3G). In addition, LIVER-ID gene repression induced by thapsigargin was blunted by hepatocyte co-treatment with cycloheximide (Appendix Fig.S25A), which, by inhibiting protein synthesis, alleviates ER stress (Harding et al., 2000). To further verify that the LIVER-ID gene repression was readily linked to ER stress, we have performed additional

experiments where hepatocytes were pre-treated with the chemical chaperone 4-phenylbutyrate (PBA). As expected, LIVER-ID gene repression was blunted by this ER stress inhibitor (Fig.2G). In addition, novel data using UPR inhibitors indicated that signaling through the different ER stress sensors contributes to LIVER-ID gene/TF repression (Fig.5 and Appendix Fig.S24). In particular, as requested by the reviewer, we have performed additional experiments using ISRIB to inhibit signaling through PERK. As suspected by the reviewer, the data indicated that this pathway contributes to ER stress-induced repression of LIVER-ID TFs, especially NR1H4 (Fig.5D-E). The conclusion that different ER stress sensors contribute to LIVER-ID gene/TF repression is consistent with previous studies indicating that the different UPR arms are interconnected and coordinately mediate ER stress effects (Brewer, 2014), as stated p.15 lines 19-22 in the discussion of the revised manuscript.

2. As mentioned above, I don't see the connection between some of the results and are at best correlative with the current provided evidences. In my view, the manuscript will be more impactful and attractive if the authors decide to remove some of the distracting data (can the manuscript be separated into 2?). It will also leave room to add key data that will directly link the different observations with strong evidence including what is described in "Major point #1".

We have significantly modified our description of the data in order to make the result section more straightforward. In particular, the data related to NFIL3 have been incorporated in a result section entitled "Acute ER stress triggers a global loss of activity of the LIVER-ID TF network and its densely co-bound CRMs". This allows to put less emphasis on this part of the manuscript and clarifies how these data relate to the central concept of our study i.e. LIVER-ID TF-mediated control of gene expression under ER stress. Indeed, we now make it clearer that these data exemplify how loss of LIVER-ID TF expression is comprised within a broader remodeling of the TF repertoire where induction of NFIL3 adds up to loss of the LIVER-ID TF HLF to further downregulate targets of the PAR-bZIP TF family (p.9 lines 1-2 and lines 24-26). We have also clarified the functional connections linking ER stress to loss of hepatic molecular identity both by adding novel results and improving the description of the data (please also see the response to Reviewer's 2 Major point 2c for additional information related to this issue). The model (Fig.8; p.51 lines 18-21) has also been revised to accommodate these changes.

3. I find it very unconventional to use the term ERS (ER stress) repeatedly to describe the upregulated transcriptional program that is modulated by the unfolded protein response (UPR). The UPR is activated by the ER stress transducers IRE1 α , PERK, ATF6. The authors should refer to the UPR through the manuscript. I think a short introduction on the UPR pathway is needed as it is central to the story. The UPR (referred to ERS by the authors) seems to be a black box even in the proposed model in Fig. 6.

We agree with the reviewer that a presentation of the ER stress sensors and the UPR was lacking. This has been added to the introduction (p.4 lines 8-15). Also, we acknowledge that using "ERS genes" to refer to genes upregulated upon stress was inappropriate. We now consistently refer to "ERS UP genes" both in the manuscript and the figures. We would like to propose to keep this nomenclature in order for the reader to better appreciate the dichotomy between upregulated and downregulated genes (i.e. "ERS UP versus ERS DOWN genes" rather than "UPR versus ERS DOWN genes"). Moreover, one potential extrapolation of our study could be that gene downregulation is an integral part of the response to ER stress and not a bystander, defining ER stress downregulated genes as part of the UPR. Notwithstanding these semantic issues, when using the "ERS UP genes" terminology for the first time in the manuscript, we now state that this refers to what is traditionally referred to as the UPR (p.5 lines 18-19). Finally, the model (Fig.8; p.51 lines 18-21) has been revised to better display how LIVER-ID gene repression is triggered by ER stress-induced signaling.

4. The detection levels of p-Y416-SRC and total SRC are quite low. The loading control is also of poor quality. There are many examples of strong p-Y416-SRC detection in the literature (PMID 15546918, 22411953, 23145001). It needs to be revised.

These immunoblots have been re-run using a more sensitive device. The new images which were obtained are shown together with their quantifications in Fig.5H. The materials and methods section has been updated accordingly (p.31 lines 7-10).

5. I find it strange to have some of the results description buried into the supplementary file (additional results). I think it should be moved to the main text unless the journal policies allow it.

This material has been moved to the main text (p7 lines 20-26, p.15 lines 24-26 and p.16 lines 1-12).

Minor points

There are some minor points that should be considered below before publication if they haven't been already addressed by the authors.

Minor Points

1. Page 5, replace "These hepatic ERS UP gens were" by "These hepatic ERS UP genes were".

The correction has been made.

2. Page 16, there is a parenthesis missing after "expression (events occurring in MPH in the absence...".

The correction has been made.

3. I dislike the use of the term "4h ERS" instead of "4h Tm". I think it is really misleading especially when put together in a figure together with inhibiting drug like MG132. Please not that by convention we use the term Tm instead of Tn for tunicamycin. This should be corrected through the manuscript and the figures.

Throughout the study, we have used both tunicamycin and thapsigargin to induce ER stress. The specific compound used in each experiment is clearly defined in the materials and methods and figure legends. We feel the use of "ERS" as a generic name in the figures better allows non-specialists to capture the main message which is conveyed by our data. Importantly, both drugs similarly induce LIVER-ID gene repression (Appendix Fig.S3G, S6E and S6G). Abbreviations commonly used in the field of ER stress (Tm for tunicamycin and Tg for thapsigargin) are now properly used in the Appendix when the two drugs are being directly compared. Moreover, based on the reviewer's comment, Fig.5 has been amended to replace "4h ERS" by "ERS (4h Tg)" and "subjected to 1h ERS" has been changed to "1h after ERS induction" when describing Fig.5F (p.12 line 15).

References

Brewer JW (2014) Regulatory crosstalk within the mammalian unfolded protein response. *Cell Mol Life Sci* 71: 1067-79

Harding HP, Zhang Y, Bertolotti A, Zeng H, Ron D (2000) Perk is essential for translational regulation and cell survival during the unfolded protein response. *Mol Cell* 5: 897-904

Reviewer #2

In the manuscript "Endoplasmic reticulum stress actively suppresses hepatic molecular identity in damaged liver", Vanessa Dubois and colleagues address a well-established phenomenon: Hepatic endoplasmic reticulum stress (ERS) represses the expression of a set of genes that determine liver functions. The analysis of published transcriptomic and genomic data is nicely combined with new, well-designed experiments whereby the authors characterize the inhibition of liver identity genes under diverse ER stress conditions (ERS). To that aim the authors take advantage of several experimental systems where ERS is induced.

From the transcriptomic analysis of thapsigargin-treated primary hepatocytes, Dubois and colleagues define sets of ERS-induced and ERS-repressed genes. In agreement with previous observations, a significant fraction of ERS downregulated genes encodes factors that support the metabolic functions of differentiated hepatocytes; these genes are identified in this manuscript as LIVER ID genes. Under non-ERS conditions, the expression of LIVER ID genes is robust and (to a large extent) hepatospecific. Epigenetically, this set of genes is featured by broad H3K4me3 histone marks at their promoters, and high levels of H3K27 acetylation, as well as by the association of a node of multiple hepatospecific transcription factors (LIVER ID TFs) that, in turn, recruit BRD4 into superenhancer domains. This work documents the molecular alterations whereby LIVER ID genes are repressed under ERS: 1) the enhancer H3K27ac mark decreases (while it increases in ERS up genes), 2) BRD4 is decommissioned from these promoters, and 3) the level of several LIVER ID TFs is reduced. Nicely enough, the analysis of ChIPseq data confirmed that robustness of LIVER ID gene expression is established by the coordinated binding of multiple liver IDs TFs to LIVER ID genes. This manuscript identifies two mechanisms, 1) the degradation of LIVER ID TFs through an SRC-dependent mechanism or 2) the inhibition of specific LIVER ID TFs by the ERS-induced transcriptional inhibitor NFIL3 in the inhibition of part of the hepatic gene expression program.

Together, the evidences presented in this manuscript integrate previous and new observations into a model that, again, is coherent with earlier published data; the novelty of some findings is relative. On the other hand, this work proposes specific molecular mechanisms that would account for LIVER ID gene repression. While these mechanisms may explain how liver transcription is tuned under ER stress, the experimental evidence supporting them is sometimes insufficient or not precise enough and it should be revised before publication.

Major points:

1. The authors document that the reduction of LIVER ID mRNAs levels under ERS is stronger than the decrease of "ubiquitous" genes (featured by the presence of broad H3K4me3 marks at their promoters) or "other genes"; thus, it is tempting to assume that the mechanism of LIVER ID gene inhibition is selective. Alternatively, the extent of transcriptional inhibition could be proportional to initial/steady-state expression levels in such a way that it could be easier to score in highly expressed (i.e. LIVER ID) genes. To rule out that possibility, the transcriptional downregulation of LIVER ID and ubiquitous genes having similar basal expression levels should be compared.

In order to define whether the extend of transcriptional repression was linked to gene basal expression levels, we performed additional analyses by plotting fold changes induced by ERS against initial gene expression levels for both LIVER-ID and UBQ genes. These new data revealed no correlation between the extend of ERS-mediated gene repression and initial gene expression levels. These data are shown in Appendix Fig.S3E (same scale was used for the x axis in the 2 plots as indicated in the figure legend) and described on p.6 lines 24-26.

2. The experiments performed to address the decommissioning of BRD4 from the LIVER ID genes convincingly show that, under ER stress, the decreased association of BRD4 with LIVER ID promoters

correlates with lower H3K27 acetylation of these genes. Based on these observations, as well as on the ChIP-PCR analysis of BRD4 association to LIVER ID genes (Figure 2G), this manuscript proposes that BRD4 is redistributed from LIVER ID to ERS genes where it could convey higher transcription rates. Unlike LIVER ID genes, ERS genes are not really sensitive to BRD4 inhibition and therefore the redistribution of BRD4 may not be relevant for ERS transcriptional activation. While many of the evidences supporting the correlation between LIVER ID binding, BRD4 recruitment and H3K27 acetylation are in good agreement with the main conclusions of this work, there are may instances where more/better evidence is needed.

a. In the Western blot of Fig. 2F BRD4 protein levels could be reduced by ER stress; the long band seem to be saturated in the control samples samples, but ERS may reduce the signal of the long band. This ERS-induced reduction seems a bit more clear in the MZ1+ERS samples (when compared to cells treated with MZ1 alone).

The image originally shown has been replaced in the main figure (Fig.4B; former Fig.2F) by another one from the same western blot obtained using a shorter exposure time. This image shows that ERS does not lead to reduced BRD4 levels. Additional independent experiments led to the same conclusion (Appendix Fig.S10C). Quantifications of BRD4 protein levels from these different assays are now shown in a novel bar graph in Fig.4B.

b. The ChIP-PCR analysis shown in Fig. 2G represents genomic domains corresponding to 8 ERS genes and 5 LIVER-ID TF genes, which were analyzed in Figs. S7 and S8. Given the low number of genes/genomic locations analyzed, it would be good to show the data corresponding to each individual ChIP-PCR (as Supplementary data) rather than an average of all measurements.

Results for the individual regions are now provided in Appendix Fig.S10B. The box plots in Fig.4A (former Fig.2G), while displaying the heterogeneity in the response of individual regions, allow to capture the overall pattern of changes in BRD4 and H3K27ac levels at the ERS UP and LIVER-ID genes.

c. The correlation between LIVER ID TF binding to LIVER ID genes, the extent of H3K27 acetylation and the recruitment of BRD4 depicts the molecular features of LIVER ID genes (and the loss of these features characterizes their inhibition); but, how are these molecular features functionally related with each other in the context of ERS-dependent transcriptional inhibition? Can we establish their contribution to LIVER ID downregulation more precisely? For instance, Trichostatin A suppresses the inhibition of LIVER ID genes under ERS, but does it prevent the rapid degradation of LIVER ID TFs? Is BRD4 inhibition sufficient to promote LIVER ID TF degradation? How plausible is it that the destabilization of LIVER ID promoters/enhancers results from the inhibition of one of the events considered in this study?

Our data firmly establish that loss of LIVER-ID TF expression and activities at hepatic CRMs is central to the partial hepatic dedifferentiation triggered by acute ERS (Fig.2 and 7). We propose that this translates into decreased H3K27ac levels and BRD4 decommissioning at super-enhancers. Indeed, LIVER-ID TFs interact with the histone acetyltransferase EP300 (Appendix Fig.S10A and Fig.S15B) (Eeckhoute et al., 2004, Kemper et al., 2009, von Meyenn et al., 2013), which is responsible for acetylation of H3K27. Moreover, the LIVER-ID TF HNF4A has recently been shown to be required for the deposition of H3K27ac at hepatic enhancers (Thakur et al., 2019). The functional connection between EP300 acetyltransferase activity and BRD4 recruitment to CRMs has been shown in (Roe et al., 2015) (Appendix Fig.S15B). The relevance of these mechanisms is demonstrated by our data using EP300 and BRD4 inhibitors, which dampen LIVER-ID basal gene expression and blunt/abolish ERS-mediated repression (Fig.4C and Appendix Fig.S11, S13-14). Among the drugs we used, JQ1 acts by inhibiting the binding of BRD4 to H3K27ac and other acetylated proteins. Altogether, this defines the direct functional connection between LIVER-ID TF and both H3K27ac and BRD4 in the establishment of

active hepatic CRMs, which is compromised upon acute ERS. The description of the data (p.10 lines 3-11 and p.11 lines 18-23) and the proposed model (Fig.8) have been modified to clarify these points. In order to better define how acute ERS interferes with hepatic CRM activities, we therefore investigated in greater details how ERS induces loss of LIVER-ID TF expression. LIVER-ID TFs form a network where auto- and cross-regulations define a positive feedback loop where they sustain their own expression to high levels in hepatocytes (Kyrmizi et al., 2006). This is accomplished through LIVER-ID TF recruitment to their own genes at BRD4 super-enhancers (Appendix Fig.S15B and S16). Our data indicate that disruption of this positive feedback loop is a primary and key event explaining the effects of acute ERS on the hepatic molecular identity (Fig.2 and 5). We propose a model where early downregulation of LIVER-ID TF protein levels (through both PERK-mediated translation inhibition and protein degradation (Fig.5 and 8)) serves as an initial trigger disrupting this positive feedback loop by disturbing BRD4 super-enhancers. In line with this proposal, we have performed additional experiments indicating that BRD4 inhibition with MZ1 affects LIVER-ID TF gene expression but does not induce early degradation/protein loss (Appendix Fig.S12). Altogether, our data support a model where acute ERS interferes with LIVER-ID TF expression by disrupting the positive feedback loops involved in sustaining the hepatic transcriptional network. This secondarily translates into compromised expression of their target LIVER-ID genes and partial hepatic dedifferentiation. The functional connections involved in hepatic CRM activities and the effects of acute ERS are summarized in a revised Fig.8.

3. The levels of some of LIVER ID transcription factors are strongly reduced by ER stress. The experimental evidence nicely documents that in thapsigargin-treated hepatocytes HNF4A, NR1H4 and FOXA2 protein levels drop even before mRNA levels change in response to ERS. The authors state that this reduction is due to the enhanced proteasomal degradation of these TFs, since addition of MG132 yields higher TF levels. However, when compared with the cells treated only with MG132, the combined thapsigargin+MG132 also reduces the levels of FOXA2 or HNF4A (and even of NR1H4), indicating that this phenomenon still occurs when proteasomes are inhibited. In the absence of a quantitation of these Western blots (that the authors should include), it seems clear that proteasomal degradation of these factors does not account for the downregulation of LIVER ID TFs observed. Similarly, the treatment of stressed cells with PP2 does not fully prevent the dampening of LIVER ID TFs.

As requested by the reviewer, we now show quantifications of the LIVER-ID TF protein levels in experiments involving MG132 and PP2 (Fig.5G and 5I). While the data indeed show that these drugs do not prevent ERS-mediated downregulation of all analyzed TFs, the results support a role for SRC-mediated proteasomal degradation of HNF4A upon acute ERS (Fig.5D-I). Interestingly, inhibiting SRC was sufficient to blunt secondary LIVER-ID TF/gene transcriptional repression (Fig.5J). Hence, while these results indicated a role for SRC-mediated LIVER-ID TF downregulation, they indeed suggested that additional mechanisms may be involved. We have therefore performed additional experiments towards this line of investigation as detailed in response to the next point.

4. Since proteasomal degradation may not be the main (or the only) driver of LIVER ID TF decay, alternative mechanisms should be considered. For instance, the translational inhibition imposed by the ERS transducer PERK could prevent HNF4A/FOXA2 translation and thereby contribute to their downregulation. Pharmacological inhibition of PERK signaling using ISRIB or GSK2606414 could address this possibility.

As suggested by the reviewer, we have performed additional experiments where PERK signaling was inhibited by ISRIB. These new data indicate that PERK signaling indeed also contributes to the early loss of LIVER-ID TF since its inhibition protected from the loss of NR1H4 (Fig.5D-E). Overall, as now acknowledged p.12 lines 19-20 and p.16 lines 13-15, our study indicates that early repression of LIVER-ID TFs triggered by ERS involves the concomitant effects of different signaling pathways. In line, we acknowledge in the discussion that additional mechanisms, in addition to those described in this study,

most probably contribute to LIVER-ID TF rapid downregulation. The model (Fig.8; p.51 lines 18-21) has been modified to incorporate these novel findings.

5. Along with this idea, it is somewhat surprising that the potential link between the three independent UPR mechanisms and the inhibition of LIVER ID genes has not been explored to some detail in this work. In fact, SRC-dependent degradation of TFs depends on IRE1. NFIL3 has been identified as a potential ATF6 target gene, while CHOP expression is controlled by the three UPR signaling mechanisms but maybe most importantly by PERK. Therefore, it would be in the scope of this work to determine how/which UPR branch (if any) mediates the downregulation of LIVER ID genes.

Based on the reviewer's recommendation, we have performed a series of additional experiments using inhibitors of the 3 arms of the UPR (Fig.5 and Appendix Fig.S24). Together with the data obtained using the SRC inhibitor PP2 (Fig.5F and 5H-J), the results indicate that LIVER-ID gene repression and *Nfil3* induction could not be ascribed to a single sensor/signaling pathway. The conclusion that different ERS sensors contributes to LIVER-ID gene/TF repression is consistent with previous studies indicating that the different UPR arms are interconnected and coordinately mediate ERS effects (Brewer, 2014), as stated p.15 lines 19-22 in the revised manuscript.

6. Is sepsis a good surrogate of ER stress? How specific is the link between ERS and LIVER ID downregulation? Elegantly, the authors demonstrate ERS is resolved after hepatectomy in wild-type but not in ATF6-deficient mice. Unresolved ER stress leads to a prolonged inhibition of LIVER ID genes. This observation establishes a nice functional correlation between UPR/ERS and liver ID gene expression.

But, would any type of liver damage cause the same downregulation of liver identity genes? Is sepsis causing this "loss of identity" through ERS or could it be linked to a more unspecific, general stress response? PBA or TUDCA have been shown to alleviate ER stress signaling in the liver of mice; while the mechanism by which these drugs relieve ER stress is still unclear, the authors may use these compounds to evaluate their capacity to prevent (or diminish) LIVER ID downregulation in sepsis models. Other than that, it would be interesting to test if hepatic insults not linked to ERS could promote the decreased expression of liver identity genes.

As discussed at the end of the manuscript (p.19 lines 3-8) and highlighted by our analyses (Fig.6A-D and Appendix Fig.S21A-B), sepsis triggers profound transcriptomic alterations that are, most probably, not solely driven by ERS. Therefore, we cannot rule out that additional pathways contribute to the described loss of hepatic identity in this context. However, based on the reviewer's recommendation, we have used TUDCA to alleviate ERS signaling in septic mice. We found this allowed to prevent downregulation of LIVER-ID TF expression (Fig.6H-I and Appendix Fig.S22D-E). These novel data, described p.13 lines 15-21, support the contribution of ERS to loss of hepatic identity in sepsis. Moreover, as suggested by the reviewer, we mined transcriptomic data from additional mouse liver injury models (Table EV7) which also revealed concomitant induction of ERS handling genes and loss of LIVER-ID gene expression (Appendix Fig.S20). Although these gene expression modulations do not occur to the same extent in all analyzed experimental conditions, additional data mining allowed to indicate that liver injury models considered in our study are associated with partial hepatic dedifferentiation (Fig.6D). While, of course, this does not imply that any type of liver injury would be characterized by loss of hepatic identity, our findings demonstrate that this is not specific to sepsis. These novel data are described on p.13 lines 3-9.

Minor points:

1. In Fig. 1C, the authors claim that PHx and ERS are associated with more complex and widespread transcriptomic alterations than, for instance, the physiological fast/fed metabolic shift. This comparative analysis was performed to determine if downregulation of the hepatic gene expression program under acute stress might occur as a consequence of the exceptional requirement for induction

of novel genes. Although the analysis documents nicely the amplitude of transcriptomic responses under ERS, it does not address properly/solve this relevant question.

As discussed p.18 lines 9-13, our data are consistent with the recent proposal that ecosystem-like equilibriums may explain the extent of coordinated gene up and downregulation in response to (patho)physiological challenges (Silveira & Bilodeau, 2018). However, we agree with the reviewer that firmly demonstrating that this model holds true will require extensive additional work and experimental evidences. Hence, we have revised the description of Fig.1C to avoid any confusion with regards to the conclusion drawn from these analyses (p.6 lines 4-13).

2. The effect of *NFIL3* downregulation in xenobiotic metabolism genes is modest, but still significant. The authors should provide a table including the list of genes that are less downregulated by ERS in the *NFIL3* ko as well as the extent of downregulation and the functional category to which they belong.

We have added to the manuscript a novel table (Table EV4), which provides additional details regarding the transcriptomic consequences of ERS in the *NFIL3* KO mice. We now provide the list of LIVER-ID genes whose ERS-mediated repression is the most significantly blunted in *NFIL3* KO mice (from Fig.3D). Moreover, we have performed additional unbiased GO term enrichment analyses to define potential additional liver functions whose repression by ERS might be less pronounced in *NFIL3* KO mice. Several other pathways involving cytochrome P450 genes were retrieved from these novel analyses as now indicated p.9 lines 17-18 and shown in Table EV4.

3. The variability of data in Fig. 5 M is really high. It is hard to draw any conclusion from these data. Of note, *CHOP/DDIT3* expression could result from the activation of *eIF2alpha* kinases other than *PERK* (i.e. *PERK*, *HRI*) that would be very likely activated in septic livers, so *CHOP* expression is not the best surrogate of ERS/ERS responses.

Data issued from human samples are typically much more variable than those obtained from mouse models. Importantly, despite human sepsis being a heterogeneous disease, the correlation plot clearly indicates that ERS markers are overall anti-correlated with LIVER-ID gene expression in septic patients (Fig.7E). However, we nevertheless changed the conclusion drawn from the human data to acknowledge that higher *DDIT3* is suggestive of a more severe ERS and/or of activation of additional detrimental signaling pathways which would add up to ERS in patients with liver dysfunction (p.15 lines 8-9).

References

Brewer JW (2014) Regulatory crosstalk within the mammalian unfolded protein response. *Cell Mol Life Sci* 71: 1067-79

Eeckhoutte J, Formstecher P, Laine B (2004) Hepatocyte nuclear factor 4alpha enhances the hepatocyte nuclear factor 1alpha-mediated activation of transcription. *Nucleic Acids Res* 32: 2586-93

Kemper JK, Xiao Z, Ponugoti B, Miao J, Fang S, Kanamaluru D, Tsang S, Wu SY, Chiang CM, Veenstra TD (2009) FXR acetylation is normally dynamically regulated by p300 and SIRT1 but constitutively elevated in metabolic disease states. *Cell Metab* 10: 392-404

Kyrmizi I, Hatzis P, Katrakili N, Tronche F, Gonzalez FJ, Talianidis I (2006) Plasticity and expanding complexity of the hepatic transcription factor network during liver development. *Genes Dev* 20: 2293-305

Roe JS, Mercan F, Rivera K, Pappin DJ, Vakoc CR (2015) BET Bromodomain Inhibition Suppresses the Function of Hematopoietic Transcription Factors in Acute Myeloid Leukemia. *Mol Cell* 58: 1028-39

Silveira MAD, Bilodeau S (2018) Defining the Transcriptional Ecosystem. *Mol Cell* 72: 920-924

Thakur A, Wong JCH, Wang EY, Lotto J, Kim D, Cheng JC, Mingay M, Cullum R, Moudgil V, Ahmed N, Tsai SH, Wei W, Walsh CP, Stephan T, Bilenky M, Fuglerud BM, Karimi MM, Gonzalez FJ, Hirst M, Hoodless PA (2019) Hepatocyte Nuclear Factor 4-Alpha Is Essential for the Active Epigenetic State at Enhancers in Mouse Liver. *Hepatology* 70: 1360-1376

von Meyenn F, Porstmann T, Gasser E, Selevsek N, Schmidt A, Aebersold R, Stoffel M (2013) Glucagon-induced acetylation of Foxa2 regulates hepatic lipid metabolism. *Cell Metab* 17: 436-47

Reviewer #3

In this paper Dubois et al describe multiple mechanisms responsible for endoplasmic reticulum stress-mediated gene expression changes in the liver. The authors compared hepatic gene expression patterns in livers following partial hepatectomy, in primary hepatocytes treated with endoplasmic reticulum stress-inducing agents as well as in KO mice lacking NFIL3 or HNF4 and in livers from mouse sepsis models.

The main conclusion of the study is that liver injury-mediated downregulation of hepatocyte identity genes involves reduction of key hepatic transcription factor protein levels, via increased degradation and via decreased expression. It is demonstrated that the effect on protein degradation involves Src3 kinase activation, while the transcriptional effect is mediated by decommissioning Brd4 at superenhancers that are densely co-bound by hepatic TFs. The above findings are novel and represent a significant advance to our current knowledge. They are based on a comprehensive analysis of existing and newly generated ChIP-seq and expression profiling data sets. The analysis seems very thorough, in most cases involving parallel, independent approaches, which strengthen the conclusions. The main mechanistic findings are also supported by the analyses of data from mouse models where sepsis is induced and from livers from deceased septic patients. The study represents an important reference work, which is expected to be heavily used by other investigators.

Points to be considered before acceptance:

1. The regulatory role of NFIL3 induction in the mechanism of ERS-mediated repression of hepatic genes, seems to be overemphasized. While the presented data showing that NFIL3 expression is increased and that the expression of some of the ERS-repressed genes are less affected in NFIL3 KO mice are clear, the overall extent of changes seem quite marginal, arguing against of a major role of NFIL3 induction in the observed effects. This part of the results does not substantially affect the story and could either be removed or NFIL3 should be discussed as a minor potential contributor in the mechanism.

Our data clearly indicate that ERS and acute liver injury are associated with a switch in expression of the PAR-bZIP TF family. This includes decreased expression of the transcriptional activators of this TF family, notably the LIVER-ID TF HLF. At the same time, expression of the transcriptional repressor NFIL3 is induced. Interfering with expression of the repressor NFIL3 using KO mice allowed us to verify that modulation of the PAR-bZIP TF family activities contributes to ERS-induced LIVER-ID gene repression. Please note that repression of the LIVER-ID TF HLF (together with TEF and DBP) still occurs in NFIL3 KO mice (Appendix Fig.S8A), which most probably explains why interfering with expression of this repressor only blunts repression of their target genes. We have revised the description of these data to indicate that they reveal how loss of a LIVER-ID TF (HLF) combines with a broader modulation of its TF family member activities upon ERS (p.9 lines 1-2 and lines 24-26). These modifications, which allow to put less emphasis on the specific contribution of NFIL3 as requested by the reviewer, better connect this section with the other data presented in the manuscript. The role of NFIL3 is also less emphasized in the revised model (Fig.8).

2. The analysis in Figure 5D showing correlation of gene expression changes in ERS induced primary hepatocytes and septic livers is clear. It would be informative to show the number of overlapping genes in venn diagrams.

Based on the reviewer's recommendation, we have added a Venn diagram (Appendix Fig.S21A), which shows that overall around 2/3 of ERS-regulated genes are also significantly modulated in septic livers.

3. It is known and also shown in Figure 1 that following ERS induction or other liver injuries, hepatic gene expression patterns after the initial drop recover to normal. The mechanism of recovery should be more explicitly discussed in the text by explaining the scheme of Figure 6.

As suggested, we have added a discussion of the mechanisms allowing for hepatic molecular identity to be re-established in relation to Fig.8 (former Fig.6) (p.18 lines 15-20).

2nd Editorial Decision

7th April 2020

Thank you again for sending us your revised manuscript. I apologise once more for the delay in sending you a decision. We have now heard back from the reviewers who were asked to evaluate your study. As you will see below, the reviewers are now satisfied with the modifications made and are supportive of publication.

Before we formally accept the study, we would ask you to address the following editorial issues.

REFEREE REPORTS

Reviewer #1:

The authors have significantly revised the manuscript with additional experiments to address the previous weakness of linking the UPR to LIVER-ID. Their findings are novel and should be of interest to the readers of MSB.

Reviewer #2:

The quality of the revised version of this manuscript is highly improved, where most of the main caveats found in the original version are clarified. Among the main improvements of this new submission, new experimental evidence documenting the link between UPR signalling and Liver ID gene expression is included. In summary, the set of modifications in this new version yield a solid, useful contribution that integrates new and existing data to explain the mechanism/s by which liver identity genes are repressed by ER stress.

2nd Revision - authors' response

9th April 2020

The Authors have made the requested editorial changes.

Accepted

14th April 2020

Thank you again for sending us your revised manuscript. We are now satisfied with the modifications made and I am pleased to inform you that your paper has been accepted for publication.

Corresponding Author Name: Jérôme Eeckhoutte

Manuscript Number: MSB-19-9156